# Human receptive endometrial assembloid for deciphering the implantation window

**Yu Zhang[1,2,3,4†], Rusong Zhao[2,3,4,5†], Chaoyan Yang[2,3], Jinzhu Song[2,3,4], Peishu Liu[1], Yan Li[2,3], Boyang Liu[2,3,4], Tao Li[6], Changjian Yin[2,3], Minghui Lu[2,3], Zhenzhen Hou[2,3], Chuanxin Zhang[2,3], Zi-Jiang Chen[2,3,4\*], Keliang Wu[2,3,4\*], Han Zhao[2,3,4,5\*]**

[1]Department of Obstetrics and Gynecology, Qilu Hospital of Shandong University, Jinan, China; [2]State Key Laboratory of Reproductive Medicine and Offspring Health, Center for Reproductive Medicine, Institute of Women, Children and Reproductive Health, Shandong University, Jinan, China; [3]Key Laboratory of Reproductive Endocrinology (Shandong University), Ministry of Education, Jinan, China; [4]Research Unit of Gametogenesis and Health of ART-Offspring, Chinese Academy of Medical Sciences (No. 2021RU001), Jinan, China; [5]Center for Clinical Reproductive Medicine, the First Affiliated Hospital of Nanjing Medical University, Nanjing, China; [6]Department of Obstetrics and Gynecology, Shandong Provincial Hospital, Shandong First Medical University, Jinan, China

*For correspondence:
chenzijiang@hotmail.com (Z-JC);
wukeliang_527@163.com (KW);
hanzh80@sdu.edu.cn (HZ)

†These authors contributed equally to this work

## eLife Assessment

This **important** study reports an endometrial organoid culture system mimicking the window of implantation. The evidence supporting the conclusion drawn is **convincing**. The data will be of interest to embryologists and investigators working on reproductive biology and medicine.

**Abstract** Human endometrial receptivity is a critical determinant of pregnancy success; however, in vivo studies of its features and regulation are particularly challenging due to ethical restrictions. Recently, the development of human endometrial assembloids has provided a powerful model to investigate this intricate biological process. In this study, we established a specialized human window-of-implantation (WOI) endometrial assembloid system that mimics the in vivo receptive endometrium. It not only reproduces the structural attributes of pinopodes and cilia, but also molecular characteristics of mid-secretory endometrium. Furthermore, the WOI endometrial assembloid exhibits hormone responsiveness, an energy metabolism profile characterized by larger and functionally enhanced mitochondria, increased ciliary assembly and motility, and epithelial-mesenchymal transition (EMT), as well as promising potential for embryo implantation. As such, WOI assembloids hold great promise as a platform to unravel the intricate mechanisms governing the regulation of endometrial receptivity, maternal-fetal interactions, and associated pathologies, ultimately driving impactful advancements in the field.

## Introduction

The human endometrium, a complex tissue comprising diverse cell types, undergoes shedding (menstrual phase), regeneration (proliferative phase), and differentiation (secretory phase) under the coordinated regulation of estrogen and progesterone throughout the menstrual cycle (*Wang et al., 2020b*). A brief interval, termed the 'window-of-implantation (WOI)' or mid-secretory phase, permits embryo implantation into the endometrium (*Wang et al., 2020b*). The endometrium at the

site of embryo implantation harbors a heterogeneous population of cells, including ciliated epithelial cells, decidualized stromal cells, and immune cells. Cyclic endometrial changes analogous to those observed in humans are exclusive to apes, Old World monkeys, molossid bats, and spiny mice, but are absent in conventional laboratory mice (*Wang et al., 2020b*). This renders typical mouse models inadequate for accurately recapitulating human endometrium. Existing endometrial cell lines, such as Ishikawa and HEEC cells, are composed of a single cell type and thus fail to reproduce the intricate physiological architecture and function of the endometrium.

Replicating and reconstructing human organs has become indispensable for elucidating tissue physiology and function. Organoid, a self-assembled 3D structure, closely resembles in vivo tissue or organ (*Turco et al., 2017*). They offer high expansibility, alongside conserved phenotypic and functional properties, emerging as powerful tools for investigating tissue physiology and disease (*Boretto et al., 2017*). In 2017, the first long-term and hormone-responsive human endometrial organoid was derived from adult stem cells in endometrial biopsies (*Turco et al., 2017*; *Boretto et al., 2017*). Based on this, Margherita Y. Turco et al. generated endometrial organoids from menstrual effluent using non-invasive methods (*Cindrova-Davies et al., 2021*), while Takahiro Arima et al. engineered polarity-reversed organoids to study embryo implantation (*Shibata et al., 2024*). Beyond adult stem cells, pluripotent stem cells were also induced to endometrial stromal fibroblasts and epithelium and then cocultured to form organoids (*Cheung et al., 2021*; *Gong et al., 2022*), which exhibited vigorous proliferative capacity but lacked immune cells and other components of the microenvironment. Several studies have additionally incorporated immune cells into endometrial organoid co-culture systems (*Dolat and Valdivia, 2021*). To recapitulate the full spectrum of human endometrial cell types in vitro, endometrial assembloids have evolved from epithelial organoids (*Fitzgerald et al., 2019*), to assemblies of epithelial and stromal cells (*Wiwatpanit et al., 2020*; *Rawlings et al., 2021*) and then to stem cell-laden 3D artificial endometrium (*Park et al., 2021*; *Park et al., 2023*), which were solely closer but not completely identical to the endometrium. Additionally, pathological endometrial organoid models have been established for conditions like endometriosis, endometrial hyperplasia, Lynch syndrome, and endometrial cancer (*Boretto et al., 2019*). These organoids faithfully simulated endometrial morphology, hormone responsiveness, and physiological and pathological processes in vitro (*Boretto et al., 2017*; *Fitzgerald et al., 2019*; *Haider et al., 2019*; *Cochrane et al., 2020*; *Bui et al., 2020*), facilitating the study of physiological phenomena (*Haider et al., 2019*; *Fitzgerald et al., 2023*; *Kirkwood et al., 2022*), pathogenic mechanisms (*Cochrane et al., 2020*), and drug screening (*Boretto et al., 2019*).

Although various regulators are implicated in the WOI, our understanding of embryo implantation during this period remains limited by ethical constraints and a paucity of in vitro receptive endometrial models. The transition from proliferative to receptive endometrium involves dynamic processes, including decidualization, epithelial-mesenchymal transition (EMT), and ciliated epithelial development (*Wang et al., 2020b*; *Paule et al., 2021*), which most in vitro models have yet to fully recapitulate. Here, we established a hormone-regulated receptive endometrial assembloid system that recapitulates in vivo WOI features, providing a platform to study physiological/pathological endometrial function and maternal–fetal interactions.

## Results

### Developing receptive endometrial assembloids in vitro

To generate endometrial assembloids in vitro, pre-receptive endometrium from reproductive-age women was dissociated into single cells or small cell masses. These cells then self-assembled into assembloids induced by various small molecules, such as Noggin, EGF, FGF2, WNT-3A, and R-Spondin1 in expansion medium (ExM) (*Figure 1A,B*, *Figure 1—figure supplement 1A*). The assembloids derived from the first three generations are used for experiments (*Figure 1—figure supplement 1B*). The endometrial assembloids comprised vesicle-like glands and fibrous stromal cells (*Figure 1—figure supplement 1C*, *Figure 1—videos 1–4*). The epithelium marker E-cadherin, stromal cell marker vimentin, and endometrial gland marker FOXA2 were confirmed in cultured assembloid, which exhibited morphological similarity to the endometrium (*Figure 1—figure supplement 1D*). Moreover, the endometrial assembloids exhibited substantial expression of the proliferation marker Ki67,

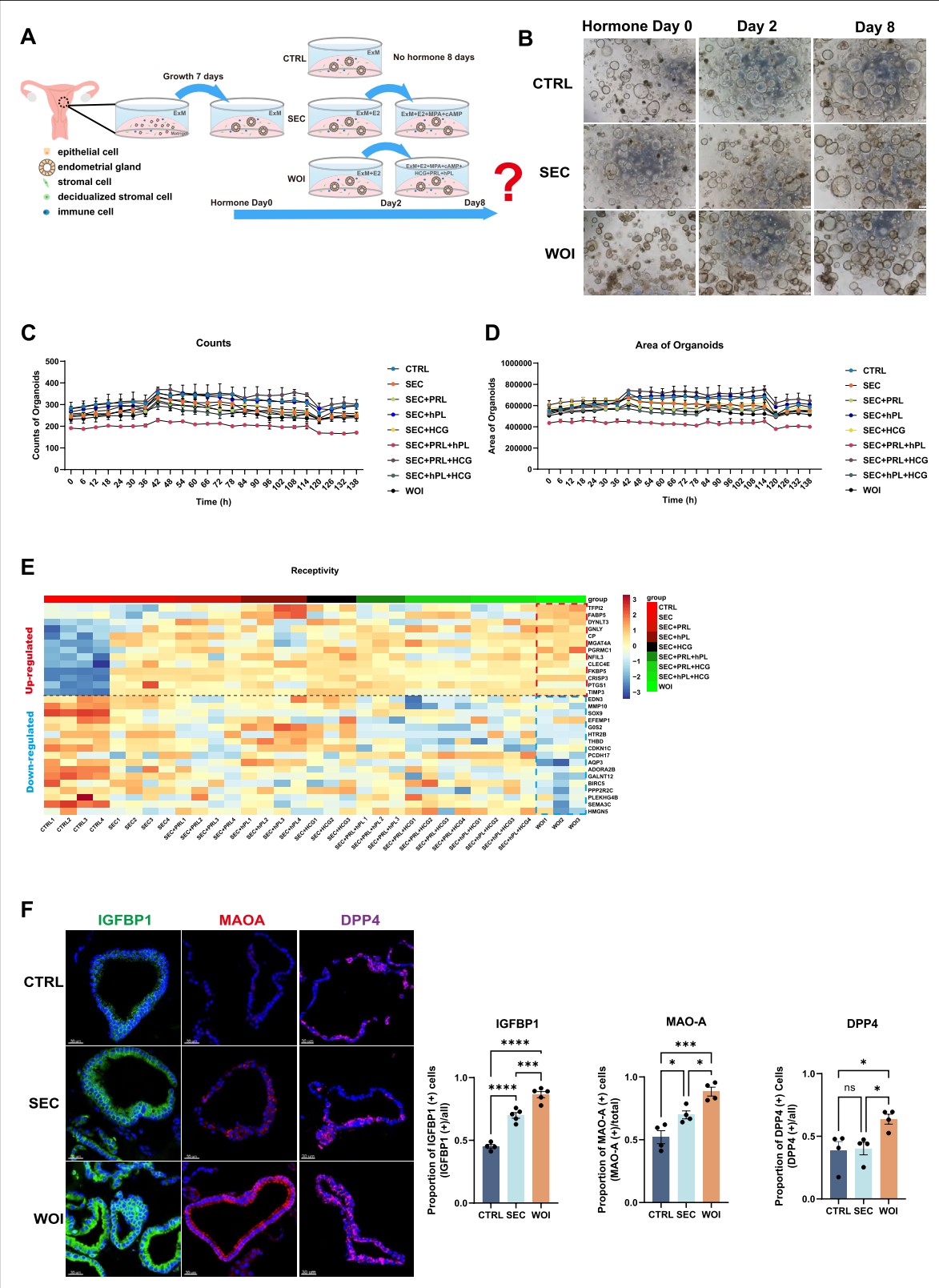

**Figure 1.** Developing receptive endometrial assembloids in vitro. (**A**) Human endometrial assembloids constructed from adult stem cells were treated with expansion medium (ExM) (CTRL) or subjected to hormonal stimulation. Timeline of endometrial assembloid cultured by ExM (CTRL), ovarian steroid hormones simulating secretory phase (SEC), ovarian steroid hormones combining PRL and placental hormones to mimic the window of implantation (WOI). (**B**) Endometrial assembloids from the CTRL, SEC, and WOI groups, which were subjected to hormone treatment on Days 0, 2, and 8, exhibited

*Figure 1 continued on next page*

*Figure 1 continued*

comparable growth patterns throughout the culture period. Scale bar = 200 μm. (**C**) The dynamic changes of the counts of assembloids over time in each hormone regimen. (**D**) The dynamic changes of the area of assembloids over time in each hormone regimen. (**E**) Heatmap showing receptivity-related gene expression profile of assembloids in each hormone regimen. The color represents log-transformed fold change of gene expression. (**F**) Validation of receptivity markers (IGFBP1, MAOA, and DPP4) with immunofluorescence (IF) in the CTRL, SEC, and WOI endometrial assembloids in vitro. Nuclei were counterstained with DAPI. Scale bar = 30 μm. The bar chart displays the quantitative comparison of receptivity markers among three groups. *$p \leq 0.05$, **$p \leq 0.005$, ***$p \leq 0.0005$, ****$p \leq 0.0001$.n=4 (CTRL) and 5 (SEC and WOI) (IGFBP1), n=4 (MAO-A and DPP4).

The online version of this article includes the following video and figure supplement(s) for figure 1:

**Figure supplement 1.** Developing receptive endometrial assembloids in vitro.

**Figure 1—video 1.** Representative video of the endometrial glands gradually developing into a vesicular shape, and the surrounding stromal cells arranging in a fibrous pattern in the endometrial assembloids, as imaged at 100 x by time-lapse microscopy of KEYENCE BZ-X800E over 72 hr.

https://elifesciences.org/articles/90729/figures#fig1video1

**Figure 1—video 2.** Representative video of the stromal cells growing in fibrous pattern and forming an extensive network in the endometrial assembloids, as imaged at 200 x by time-lapse microscopy of KEYENCE BZ-X800E over 48 hr.

https://elifesciences.org/articles/90729/figures#fig1video2

**Figure 1—video 3.** A representative video showing that the endometrial assembloids grew and differentiated, the endometrial glands enlarged, and surrounding stromal cells formed an extensive network during the control environment (CTRL), as imaged at 100 x by time-lapse microscopy of KEYENCE BZ-X800E over 8 days.

https://elifesciences.org/articles/90729/figures#fig1video3

**Figure 1—video 4.** A representative video showing that the endometrial assembloids grew and differentiated, the endometrial glands enlarged, and surrounding stromal cells formed an extensive network during the hormone treatment (window of implantation, WOI), as imaged at 100 x by time-lapse microscopy of KEYENCE BZ-X800E over 8 days.

https://elifesciences.org/articles/90729/figures#fig1video4

with no detectable cleaved caspase-3 (apoptosis marker), indicative of strong proliferative capacity (*Figure 1—figure supplement 1E*).

Functionally, endometrial assembloids effectively secreted glycogen into the lumen, mirroring the nutrient-secreting activity of the endometrium to support embryo implantation (*Figure 1—figure supplement 1D*). Moreover, after two days of estrogen (E2) treatment followed by fourteen days of medroxyprogesterone acetate (MPA) and cAMP, the assembloids exhibited significant transcriptional upregulation of progesterone receptor (PRA/B), a modest increase in estrogen receptor α (ERα), and enhanced expression of estrogen-responsive genes *EGR1* and *OLFM4*, along with progesterone-responsive genes *PGR* and *PAEP* (*Figure 1—figure supplement 1F–G*). It reflected hormonal responsiveness analogous to that of the in vivo endometrium (*Turco et al., 2017*).

To identify hormone regimens that induce the implantation window, pregnancy-related hormones were supplemented into the culture system following 2 days of E2 priming. E2 and MPA drive the transition of endometrial assembloids to the secretory phase, while pregnancy hormones promote further differentiation. Prolactin (PRL) promotes immune regulation and angiogenesis during implantation (*Turco et al., 2017*; *Auriemma et al., 2020*). Human chorionic gonadotropin (hCG) improves endometrial thickness and receptivity (*d'Hauterive et al., 2022*; *Nwabuobi et al., 2017*). Human placental lactogen (hPL) promotes the development and function of endometrial glands (*Bazer et al., 2018*). Hormone dosages were primarily based on peri-pregnant maternal systemic or local endometrium levels (*Turco et al., 2017*). Multigroup comparison revealed similar counts, area, and average intensity of assembloids over time (*Figure 1C, D*, *Figure 1—figure supplement 1H*). However, only the final cocktail (i.e. a combination of E2, MPA, cAMP, PRL, hCG, and hPL) exhibited an endometrial receptivity-related gene expression profile, with high expression of receptivity-promoting genes and low expression of receptivity-inhibiting genes relative to other hormone formulations (*Figure 1E*). The assembloids generated with this regimen were defined as WOI assembloids (*Figure 1A*). For controls, assembloids maintained in ExM served as the 'control (CTRL)' group, while those treated with 2 days of E₂ followed by 6 days of E₂, MPA, and cAMP were induced to the secretory phase (as previously described by *Fitzgerald et al., 2019*) and designated the 'secretory (SEC)' group (*Figure 1A*). There was no significant difference in the morphology of assembloids among the three groups (*Figure 1B*), but WOI assembloids exhibited elevated expression of the receptivity markers IGFBP1, MAOA, and DPP4 (*Figure 1F*), and increased glycogen secretion (*Figure 1—figure supplement 1I*). Theoretically,

the WOI assembloids originate from the secretory phase and thus share characteristics with SEC assembloids, but crucially, they are expected to exhibit hallmark features of the mid-secretory phase.

## Receptive endometrial assembloids mimicked the implantation window endometrium

Single-cell transcriptomics analysis, with reference to CellMarker, PanglaoDB, Human Cell Atlas, Human Cell Landscape, and scRNASeqDB, and prior endometrium-related studies (*Wang et al., 2020b*; *Fitzgerald et al., 2019*; *Rawlings et al., 2021*; *Garcia-Alonso et al., 2021*), identified the presence of epithelial, stromal, and immune cells within WOI assembloids (*Figure 2A*, *Figure 2— figure supplement 1A,B*). Comparative analysis of scRNA-seq data from our assembloids and mid-secretory endometrial tissue (as described by Stephen R. Quake et al. in 2020 *Wang et al., 2020b*) and Garcia-Alonso in 2021 (*Garcia-Alonso et al., 2021*) revealed that WOI assembloids exhibited similarities to the mid-secretory endometrium in terms of glandular epithelium, ciliated epithelium, and stromal cells (*Figure 2A*, *Figure 2—figure supplement 1C-F*). The morphology of immune and stromal cells was visualized via 3D clearing staining and light sheet microscopy imaging, with vimentin labeling stromal cells (Vimentin$^+$ or Vimentin$^+$ F-actin$^+$), CD45 and CD44 indicating immune cells, and FOXA2 identifying glands (*Figure 2B–D*). Furthermore, flow cytometry was employed to validate the presence and subset composition of immune cells. White blood cells (WBC) were identified as CD45$^+$ cells, with T cells, macrophages and NK cells characterized as CD45$^+$CD3$^+$, CD45$^+$CD68$^+$CD11b$^+$, and CD56$^+$CD16$^-$ cells, respectively (*Figure 2E*).

WOI assembloids displayed characteristic features of the receptive endometrium. On the one hand, they secreted significantly higher levels of glycogen into the lumen compared to other groups (*Figure 1—figure supplement 1I*). On the other hand, they possessed various characteristic micro-structures of the implantation window, including elongated microvilli and increased glycogen, pinopodes, and especially cilia (*Figure 2F*).

We next focused on transcriptional profiling and regulation during the mid-secretory phase, as this phase marks the implantation window and involves substantial transcriptional alterations (*Figure 2G–H*). For this analysis, we referred to scRNA-seq data of the mid-secretory phase from Stephen R. Quake 2020 (*Wang et al., 2020b*). Pathways related to mitochondrial energy metabolism and cell adhesion were significantly upregulated in both WOI assembloid and mid-secretory endometrium, relative to CTRL and SEC assembloids (*Figure 2G*). Furthermore, key transcription factors (TFs) associated with implantation—identified in secretory epithelial cells and EMT-derived cells— exhibited highly conserved expression patterns between WOI assembloids and the mid-secretory endometrium. Specifically, secretory epithelium shared comparable TFs linked to hypoxia response (e.g. HIF1A *Martínez-Aguilar et al., 2021*), embryo implantation (e.g. FBLN1 *Fatmous et al., 2022*), lipid metabolism (e.g. VMP1 *Jiang et al., 2022*), cell migration, and cell junction (e.g. TJP1 *Kuo et al., 2021*; *Figure 2H*). Similarly, EMT-derived cells also expressed conserved TFs involved in endometrial decidualization (e.g. S100A10 *Bissonnette et al., 2016*), EMT (e.g. FAT1 *Srivastava et al., 2018* and FOXF2 *Liu et al., 2020*), and receptivity (e.g. NEAT1 *Geng et al., 2022*, SERPINB9 *Vargas et al., 2012*, SOX17 *Kinnear et al., 2019*, and SOX4 *Huang et al., 2021*; *Figure 2H*).

To further validate the receptive phenotype of WOI assembloids, we performed Endometrial Receptivity Analysis (ERA), a gene expression profiling-based diagnostic assay that integrates high-throughput sequencing and machine learning to quantify the expression of endometrial receptivity-associated genes (*He et al., 2021*). Currently employed in clinical practice to assess endometrial receptivity and guide personalized embryo transfer, ERA revealed that WOI assembloids derived from pre-receptive endometrial tissue successfully transitioned to a receptive state (*Figure 1—figure supplement 1J*).

Collectively, these findings confirm that WOI assembloids closely recapitulate the structural and molecular characteristics of the in vivo endometrium during the implantation window.

## Receptive endometrial assembloids recapitulate WOI-associated hormone response

To characterize the biological hallmarks associated with the implantation window, we performed integrated transcriptomic and proteomic profiling of WOI assembloids, with CTRL and SEC assembloids (*Figure 3A*, *Figure 3—figure supplement 1A-E*).

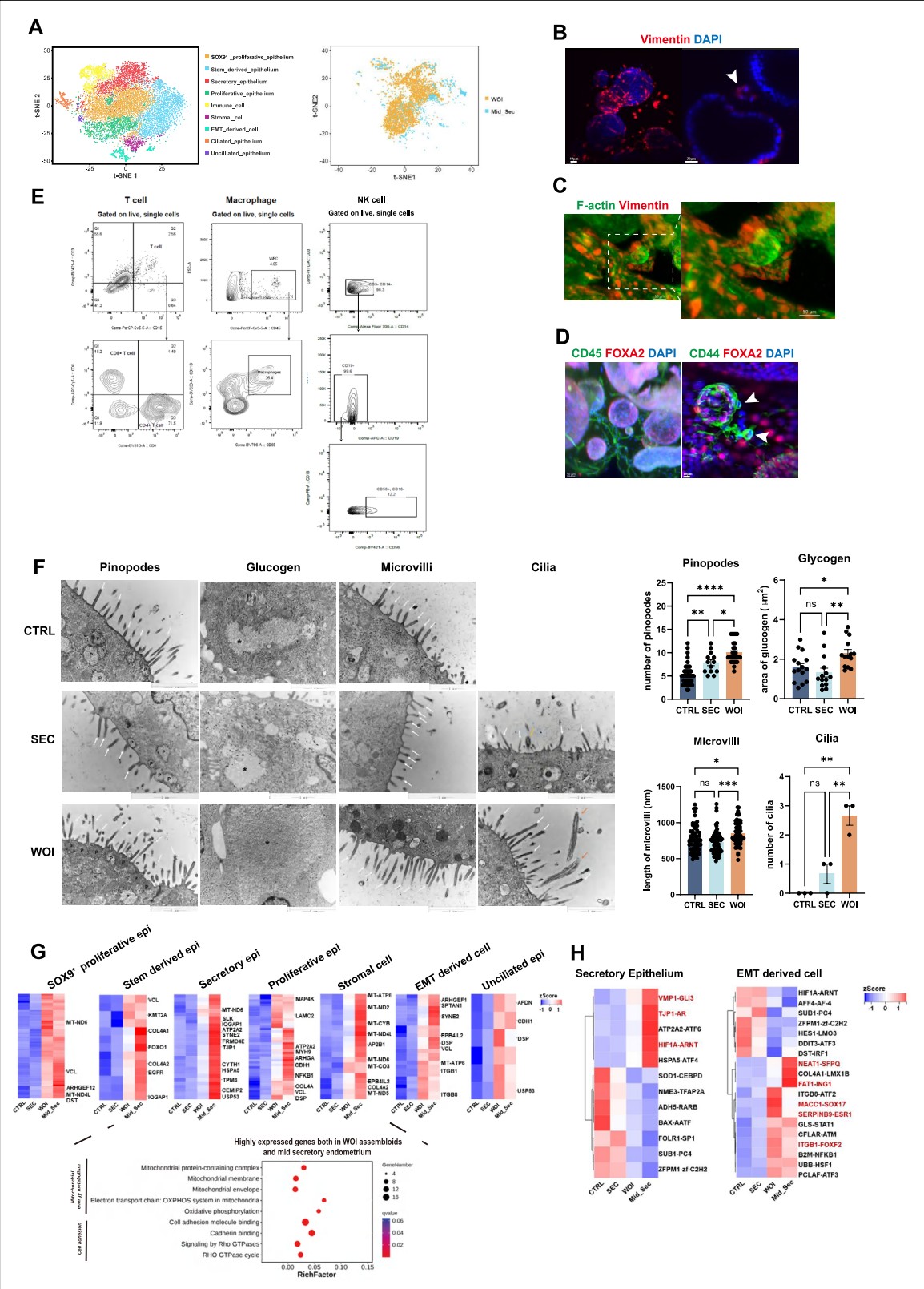

**Figure 2.** Receptive endometrial assembloids mimicked the implantation window endometrium. (**A**) T-SNE plot of scRNA-seq data from three individual endometrial assembloids of the CTRL, secretory (SEC), and window of implantation (WOI) groups (left). T-SNE plot of combined scRNA-seq data from the three kinds of assembloids and mid-secretory endometrium (right). (**B**) Exhibition of stromal cells marked by vimentin of CTRL assembloid through whole-mount clearing, immunostaining, and light sheet microscopy imaging. Nuclei were counterstained with DAPI. The arrowhead indicates stromal

*Figure 2 continued on next page*

*Figure 2 continued*

cells. Scale bar = 40 µm (left), Scale bar = 30 µm (right). (**C**) Whole-mount immunofluorescence showed that Vimentin⁺ F-actin⁺ cells (stromal cells) were arranged around the glandular spheres that were only F-actin+. Scale bar = 50 µm. (**D**) Exhibition of immune cell marked by CD45 and CD44, and endometrial gland marked by FOXA2 of CTRL assembloid through whole-mount clearing, immunostaining and light sheet microscopy imaging. Nuclei were counterstained with DAPI. The arrowhead indicates immune cells. Scale bar = 50 µm (left), Scale bar = 10 µm (right). (**E**) Flow cytometric analysis of T cells and macrophages in the CTRL endometrial assembloid. Gating strategy used for determining white blood cells (WBC) (CD45⁺ cells), T cells (CD45+CD3⁺ cells), and macrophages (CD45⁺CD68⁺CD11b⁺ cells). (**F**) Electron micrograph of the CTRL (top), SEC (middle), and WOI (bottom) endometrial assembloid showing pinopodes (**P**), glycogen granule (asterisk), microvilli (white arrows), and cilia (orange arrows). Scale bar = 1 µm. Quantitative comparison of pinopodes, glycogen, microvilli, and cilia in the CTRL, SEC, and WOI assembloids. *$p \leq 0.05$, **$p \leq 0.005$, ***$p \leq 0.0005$, ****$p \leq 0.0001$. $n = 5$, with 2–8 sections collected from each sample (Pinopodes). $n = 5$, with 2–3 sections collected from each sample (Glucogen). $n = 5$, with 4–7 sections collected from each sample. $n = 5$, with 10–13 sections collected from each sample (Microvilli). n=3 (Cilia). (**G**) Heatmap and bubble diagram illustrating highly expressed genes as well as GO functions enriched in both assembloids during the WOI and mid-secretory endometrial tissue in terms of SOX9+ proliferative epithelium, stem-derived epithelium, secretory epithelium, proliferative epithelium, unciliated epithelium, stromal cells and epithelial-mesenchymal transition (EMT)-derived cells. The color of the heatmap represents log-transformed fold change of gene expression. (**H**) Heatmaps showing differentially expressed transcription factors (TFs) of endometrial assembloids and endometrium in the secretory epithelium (left) and EMT-derived cells (right). The color represents log-transformed fold change of gene expression.

The online version of this article includes the following figure supplement(s) for figure 2:

**Figure supplement 1.** Various functions are performed by all kinds of cells identified with scRNA-seq.

**Figure supplement 2.** Receptive endometrial assembloids experienced epithelial-mesenchymal transition (EMT).

WOI assembloids exhibited a robust hormone response, as evidenced by upregulated PGR expression at the transcriptome level (*Figure 3B*). Integrated multi-omics analysis identified 179 genes/proteins that were significantly upregulated in WOI versus CTRL assembloids, the majority of which are implicated in estrogen signaling pathways (*Figure 3—figure supplement 1F–G*). Furthermore, immunostaining for PRA/B, which was used to quantify progesterone responsiveness, revealed the highest signal intensity in the WOI group, concurrent with upregulation of the estrogen-responsive protein OLFM4 (*Figure 3C*). FOXO1, a crucial marker of endometrial receptivity reliant on PGR signaling, was significantly elevated in WOI compared to CTRL assembloids (*Figure 3D*, *Figure 3—video 1*). These observations suggest that progesterone signaling is central to the establishment of the WOI phenotype in assembloids and collectively demonstrate that WOI assembloids mount a robust response to both estrogen and progesterone.

Hormonal stimulation induced an expansion of secretory epithelium and a concomitant reduction in proliferative epithelium in SEC and WOI assembloids, indicative of the proliferative-to-secretory phase transition (*Figure 3E*). At the single-cell level, the secretory epithelium, which is critical for the implantation window, was enriched for genes involved in cellular metabolic processes and hypoxic responses (HIF-1 signaling pathway) (*Figure 2—figure supplement 1G–H*). Compared to SEC assembloids, secretory epithelium in WOI assembloids exhibited enhanced peptide metabolism and mitochondrial energy metabolism (*Figure 3F*), which are functional adaptations that support endometrial decidualization and embryo implantation. Single-cell trajectory analysis further revealed that proliferative epithelium differentiates into secretory epithelium under the regulation of key nodal genes defining the transition between states 5 and 6, including KRT19, MALAT1, MT2A, and the RPL and RPS families (*Figure 3G*, *Figure 2—figure supplement 1J-L*). Notably, WOI assembloids displayed a more complete proliferative-to-secretory epithelial transition than SEC assembloids (*Figure 3G*).

Overall, WOI assembloids possessed the hormone response characteristic of implantation window, closely resembling the gene traits of embryo implantation.

## Receptive endometrial assembloids possess enhanced energy metabolism

WOI assembloids exhibited upregulation of monocarboxylic acid and lipid metabolism (represented by SLC25A1 *Yang et al., 2021*), and hypoxia response (represented by HIF1α *Martínez-Aguilar et al., 2021*; *Figure 3B*, *Figure 3—figure supplement 1G-I*). Likewise, the secretory epithelium, critical for the implantation window, contributed to cellular metabolic processes and HIF-1 signaling pathway response to hypoxia at the single-cell level (*Figure 2—figure supplement 1G–H*).

To further investigate this trend, the Mfuzz algorithm was utilized to analyze gene expression across these three groups, focusing on gene clusters that were progressively upregulated or downregulated.

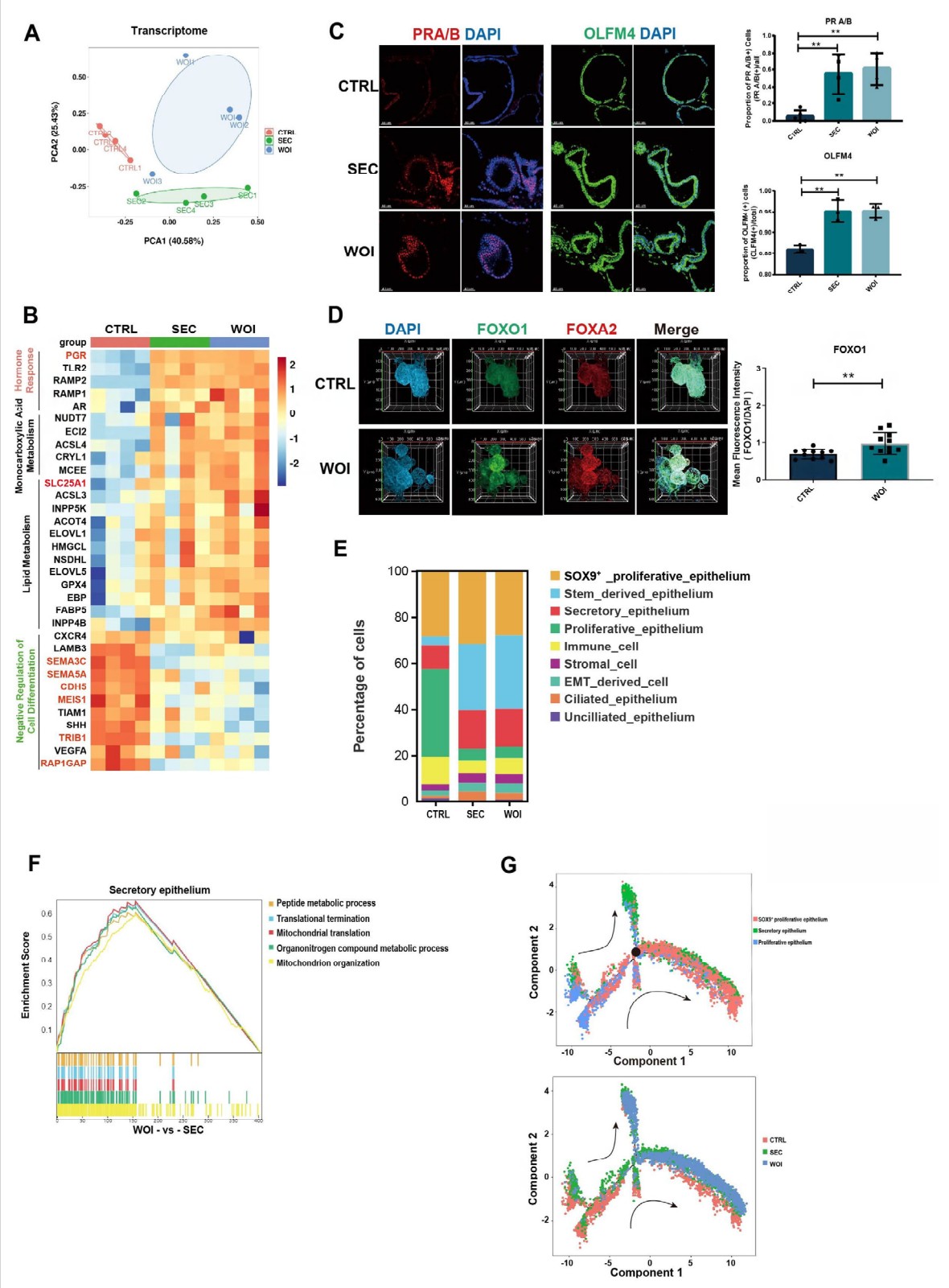

**Figure 3.** Receptive endometrial assembloids recapitulate window of implantation (WOI)-associated hormone response. (**A**) Principal component analysis (PCA) plot computed with differentially expressed genes in the bulk transcriptome of endometrial assembloids belonging to the CTRL, secretory (SEC), and WOI groups. (**B**) Heatmap showing that enrichment of differentially expressed genes for the terms of hormone response, monocarboxylic acid metabolism, lipid metabolism, and negative regulation of cell differentiation. The color represents log-transformed fold change

*Figure 3 continued on next page*

*Figure 3 continued*

of gene expression. (**C**) Responsiveness to progesterone and estrogen was evaluated by IF to PRA/B and OLFM4 with immunofluorescence (IF), respectively. Scale bar = 40 μm, **$p \leq 0.005$. n=3. (**D**) Exhibition of implantation marker (FOXO1) and endometrial gland marker (FOXA2) through combination of assembloid clearing, IF, and light sheet microscopy. Nuclei were counterstained with DAPI. **$p \leq 0.005$. *n* = 3, with 3–4 sections collected from each sample.(**E**) Bar graph exhibiting various percentages of each cell type in the three groups. (**F**) GSEA between the SEC and WOI groups for secretory epithelium. (**G**) Pseudotime trajectory showing the transformation between proliferative and secretory epithelium in the CTRL, SEC, and WOI groups. Arrows indicate the direction of the pseudotime trajectory. The black dot indicates the key differentiation node.

The online version of this article includes the following video and figure supplement(s) for figure 3:

**Figure supplement 1.** Comparisons between CTRL, secretory (SEC), and window of implantation (WOI) assembloids at the level of transcriptome and proteome.

**Figure 3—video 1.** Representative video showing that three-dimensional imaging of the chemically cleared CTRL and window of implantation (WOI) assembloids, as imaged by light-sheet microscope to detect the expression of FOXO1 and FOXA2.

https://elifesciences.org/articles/90729/figures#fig3video1

Mitochondrial-related genes were found to exhibit the highest expression levels in WOI endometrial assembloids (*Figure 4A*). Concordantly, at the protein level, WOI endometrial assembloids maintained significantly higher expression of mitochondrial proteins compared to SEC assembloids (*Figure 4B*). TEM analysis further revealed that WOI endometrial assembloids possessed the largest average mitochondrial area among the three groups (*Figure 4C*). The expression of mitochondrial-related genes increased from CTRL to SEC to WOI assembloids, with COA1 ensuring proper nuclear-mitochondrial connection (*Wang et al., 2020a*), OXA1L promoting mitochondrial translation (*Itoh et al., 2021*), and TIMMDC1 being crucial for mitochondrial complex I assembly (*Wang et al., 2021*; *Figure 4D*). WOI assembloids notably expressed higher levels of OXA1L and TIMMDC1 than the SEC assembloids (*Figure 4D*). Furthermore, WOI assembloids produced more ATP and IL8(42) (*Figure 4E*).

Thus, compared to SEC assembloids, WOI assembloids exhibited enhanced energy metabolism, underpinned by larger, functionally competent mitochondria.

## Receptive endometrial assembloids increased the ciliary assembly and motility

Cilia are specialized structural components of the endometrium, whose growth and development, assembly, and movement are essential for establishing the endometrial implantation window and facilitating embryo implantation. Under the electron microscopy, cilia were most observed in the WOI assembloids (*Figure 2E*). Concordantly, cilia-related genes and proteins exhibited peak expression levels in WOI endometrial assembloids (*Figure 5A–B*). Transcriptomic profiling further demonstrated that, relative to SEC assembloids, WOI assembloids displayed upregulated expression of genes linked to ciliary assembly, ciliary basal body, and motile cilia, whereas genes associated with non-motile cilia were downregulated (*Figure 5A*). Key differentially expressed genes in WOI assembloids included *NEK2* (ciliary assembly), *CFAP36* (ciliary basal body), and *DNAH9* and *TPPP* (motile cilia), with NEK2 showing the most pronounced upregulation (*Figure 5B*). At the protein level, WOI assembloids also exhibited elevated expression of factors involved in ciliary assembly (TBC1D1, IFT22, IFT57), ciliary basal body biogenesis (PJA2), and motile cilia function (*Figure 5C*). Additionally, acetyl-α-tubulin (cilia marker *Haider et al., 2019*) were highly expressed in the WOI assembloids (*Figure 5D*).

Single-cell transcriptome analysis revealed hormone treatment increased ciliated epithelium and decreased unciliated epithelium in SEC and WOI groups (*Figure 3E*). Ciliated epithelium functioned in protein binding, cilium organization and assembly, while unciliated epithelium acted on actin cytoskeleton and translation (*Figure 2—figure supplement 1E–F*). Notably, WOI assembloids' ciliated epithelium additionally regulated vasculature development and displayed higher transcriptional activity than the SEC group (*Figure 5E*). The ciliated to unciliated epithelial transition occurred during the menstrual cycle, which was regulated by key genes, such as *GAS5, JUN, RPL*, and *RPS* families (*Figure 5F*, *Figure 2—figure supplement 1M-O*). Given that these functional specializations are likely mediated by intercellular crosstalk, we employed CellPhoneDB (a computational tool for predicting ligand–receptor interactions *Efremova et al., 2020*) to investigate cell–cell communication networks. Ciliated epithelium of WOI assembloids interacted with immune cells and secretory epithelium, showing enhanced invasion ability via CD74-COPA (*Li et al., 2022b*), and ROR2-WNT5A (*Li et al., 2022a*), which was validated by proximity ligation assay (PLA) (*Figure 5G–H*).

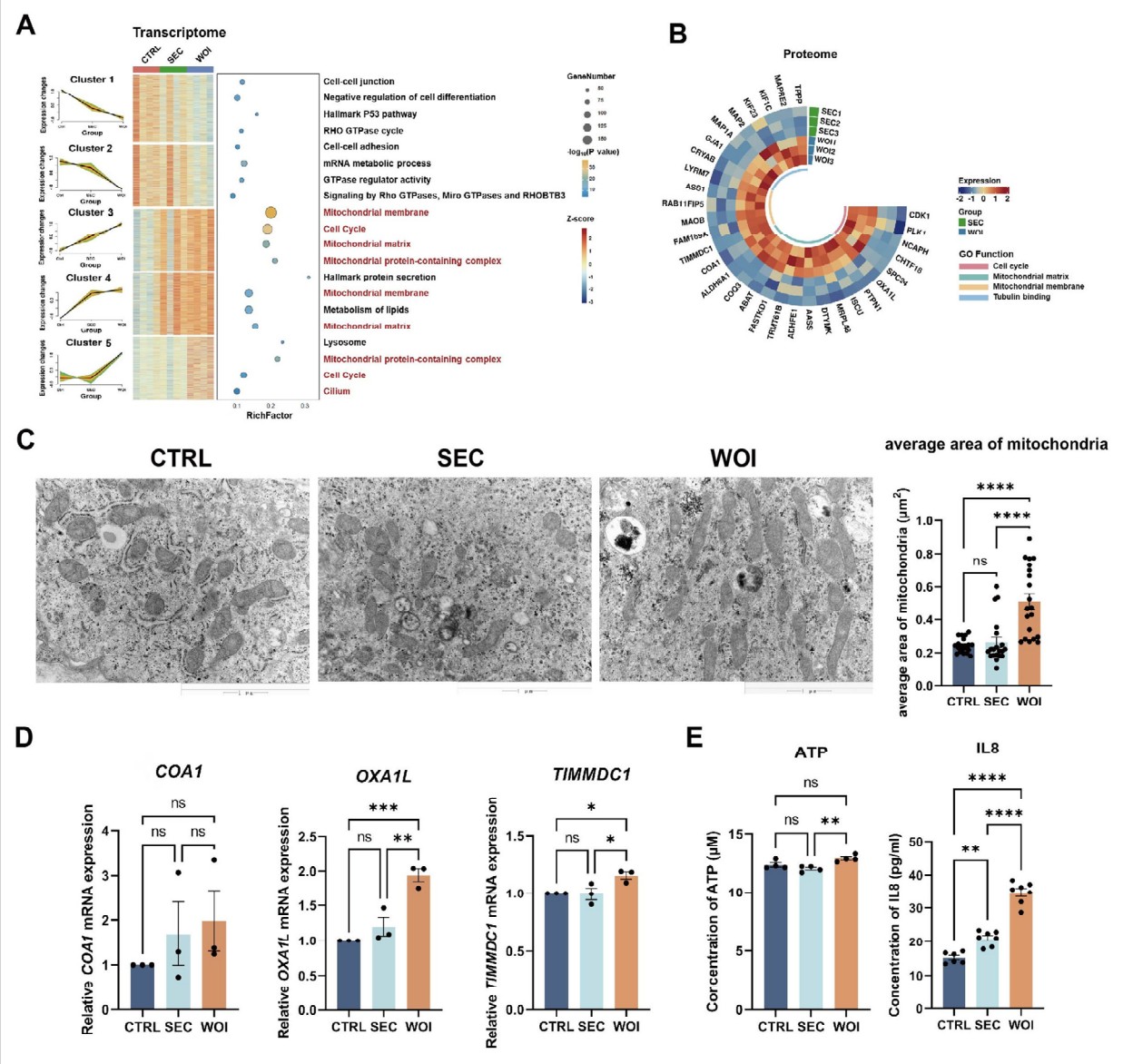

**Figure 4.** Receptive endometrial assembloids possess enhanced energy metabolism. (**A**) The Mfuzz trend analysis displayed the transcriptional variation trends of five clusters from CTRL, secretory (SEC), and window of implantation (WOI) groups (with a focus on the differences between SEC and WOI assembloids). The heatmap illustrated the corresponding gene expression profile (where color represents Z-score). The bubble plot showed the associated GO functions (with bubble size representing the number of genes and bubble color indicating the p-value). (**B**) The circular heatmap illustrated the functional differences of SEC and WOI assembloids at the proteomic level. The color represents protein expression levels, and the innermost circle color represents GO functions. (**C**) Transmission electron microscopy images displayed the mitochondrial morphology of CTRL, SEC, and WOI assembloids, along with a quantitative comparison of mitochondrial area. Scale bar = 1 μm. n = 5, with 4–6 sections collected from each sample. (**D**) RT-qPCR assessed the expression levels of mitochondrial function-related genes in the three assembloid groups. n=3. (**E**) Quantitative comparison of the concentration of ATP (left) and IL8 (right) released by CTRL, SEC, and WOI assembloids. *p<0.05, **p<0.005, ***p<0.0005, ****p<0.0001.n=4 (ATP). n=6 (CTRL) and 7 (SEC and WOI) for IL8 .

In summary, the WOI assembloids revealed the accumulated ciliated epithelium's role in preparing the implantation window, with increased ciliary assembly and motility compared to SEC assembloids.

## Receptive endometrial assembloids experienced EMT

The WOI assembloids displayed upregulated cell differentiation not only at assembloid level (*Figure 3B*) but also at cellular level, which is represented by EMT. EMT is a common and crucial biological event in the endometrium during the implantation window (*Owusu-Akyaw et al., 2019*).

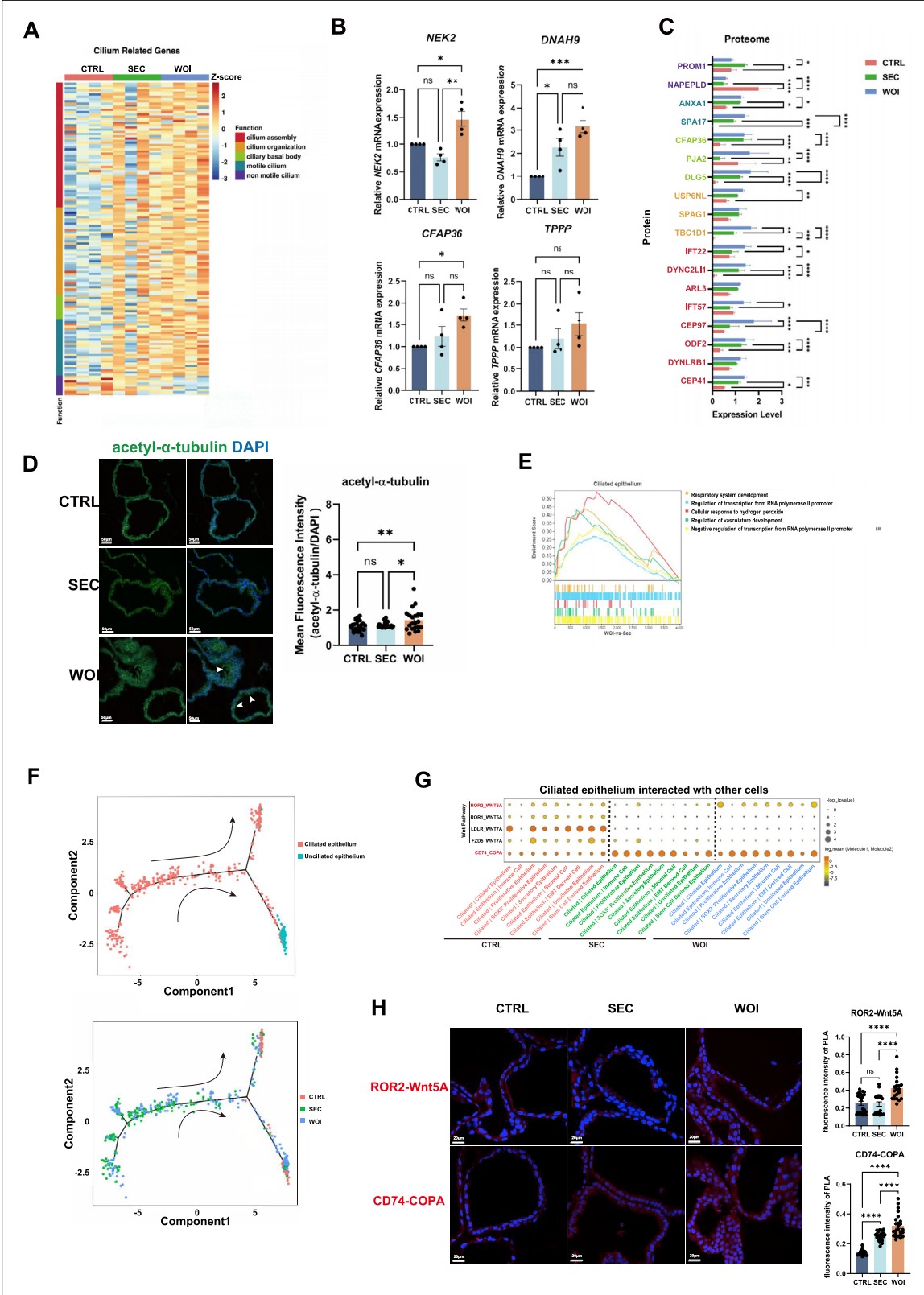

**Figure 5.** Receptive endometrial assembloids increased the ciliary assembly and motility. (**A**) The heatmap illustrated the expression of cilia-related genes across the CTRL, secretory (SEC), and window of implantation (WOI) assembloids. The color represents Z-score, while the leftmost block indicates various characteristic functions related to cilia. (**B**) RT-qPCR assessed the expression levels of cilia-related genes in the three assembloids. n=4. (**C**) The histogram showed the expression of cilia-related proteins in the three groups of assembloids. The color of the longitudinal protein names corresponds

*Figure 5 continued on next page*

*Figure 5 continued*

to the color of cilia-related functional blocks in Fig 5A. (**D**) IF analysis of cilia assembly marked by acetyl-α-tubulin. Nuclei were counterstained with DAPI. The arrowhead indicates cilia. Scale bar = 50 μm. n = 5, with 4–6 sections collected from each sample. (**E**) GSEA between the SEC and WOI groups for ciliated epithelium. (**F**) Pseudotime trajectory showing the transformation between ciliated and unciliated epithelium in the CTRL, SEC, and WOI groups. Arrows indicate the direction of the pseudotime trajectory. (**G**) Dot plots demonstrating the Cellphone DB analysis of relevant receptors and ligands of ciliated epithelium with other cell types. The size of the dot represents the level of significance. The color of the dot indicates the mean of the average expression level of interacting molecule 1 in ciliated epithelium and molecule 2 in other cell types. (**H**) Proximity ligation assay (PLA) validating the interactions of ROR2-Wnt5A and CD74-COPA in the CTRL, SEC, and WOI assembloids. Red signals the interaction of two proteins. Nuclei were counterstained with DAPI. Scale bar = 20 μm. *p<0.05,  **p<0.005,  ***p<0.0005,  ****p<0.0001. n = 5, with 4–5 sections collected from each sample.

During the EMT process, epithelial cells lose their epithelial characteristics while gaining migratory and invasive properties of fibroblasts. Integrated transcriptomic and proteomic analyses revealed increased EMT in WOI assembloids (*Figure 3—figure supplement 1G*).

EMT-derived cells, exhibiting gene expression patterns typical of epithelial and stromal cells, as well as EMT, are more abundant in the SEC and WOI groups and act on protein binding, cell cycle, organelle organization, and reproduction (*Figure 2—figure supplement 1E*). They performed enhanced lamellipodium-mediated cell migration, cell junction, and cytoskeleton regulation in the WOI assembloids compared to the SEC assembloids (*Figure 2—figure supplement 2A*). Single-cell trajectory analysis further revealed that WOI assembloids underwent a more complete differentiation from proliferative epithelium to EMT-derived cells than SEC assembloids, a process governed by key regulatory genes including *DOC2B*, *FXYD3*, and *LPCAT3* (*Figure 2—figure supplement 2B–D*). EMT-derived cells and epithelium coordinate functionally during the implantation window (*Figure 2—figure supplement 2E*). Specifically, EMT-derived cells highly expressed NRP1 and SLC7A1, while their receptors (SEMA3A and CSF1) were more upregulated in the epithelium of WOI versus SEC assembloids (*Figure 2—figure supplement 2E–G*). NRP1-SEMA3A has been reported to promote vascularization and respond to hypoxia (*Casazza et al., 2013*). SLC7A1 (*Wang et al., 2014*) and CSF1 (*Robertson et al., 2018*) both support receptivity establishment, embryo implantation, and development. Compared with canonical stromal cells, EMT-derived cells exhibited distinct patterns of crosstalk with epithelial or immune cells. For instance, CD44 and CD46, known for their roles in cell adhesion (*Sancakli Usta et al., 2020*) and immunoregulation (*Le Friec et al., 2012*), are highly expressed in stromal cells and EMT-derived cells, respectively, and bind separately with SPP1 (in epithelial cells) and JAG1 (in stromal cells) (*Figure 2—figure supplement 2E–G*).

In general, WOI assembloids recapitulate the EMT-driven endometrial remodeling process that underpins the transition to the implantation window.

## The receptive endometrial assembloids possess the potential for embryo implantation

To validate WOI assembloids' ability to recapitulate embryo implantation (a key biological process of receptive endometrium), we established an assembloid-based co-culture system. Given the rarity and ethical constraints associated with human embryos, we employed blastoids (corresponding to the human embryo at 6 days post-fertilization, referred to as 'Day 6') for implantation into the endometrial assembloids (*Figure 6A*). By Day 9, blastoids were observed to survive and integrate within the endometrial assembloids (*Figure 6B*). Importantly, co-cultured blastoids retained the capacity for normal tri-lineage differentiation, as evidenced by the appropriate expression of epiblast (OCT4), hypoblast (GATA6), and trophoblast (KRT18) lineage markers (*Figure 6C*). We next quantified blastoids survival rates and their rates of interaction with endometrial assembloids across the CTRL, SEC, and WOI groups. Strikingly, blastoids co-cultured with WOI assembloids exhibited significantly higher survival and interaction rates compared to CTRL and SEC assembloids, with survival rates of 66% (WOI), 19% (CTRL), and 28% (SEC), and interaction rates of 90% (WOI), 47% (CTRL), and 53% (SEC), respectively (*Figure 6D–E*). This demonstrates that WOI assembloids possess a markedly enhanced capacity to support blastoids survival and integration, highlighting their functional relevance to the receptive endometrium.

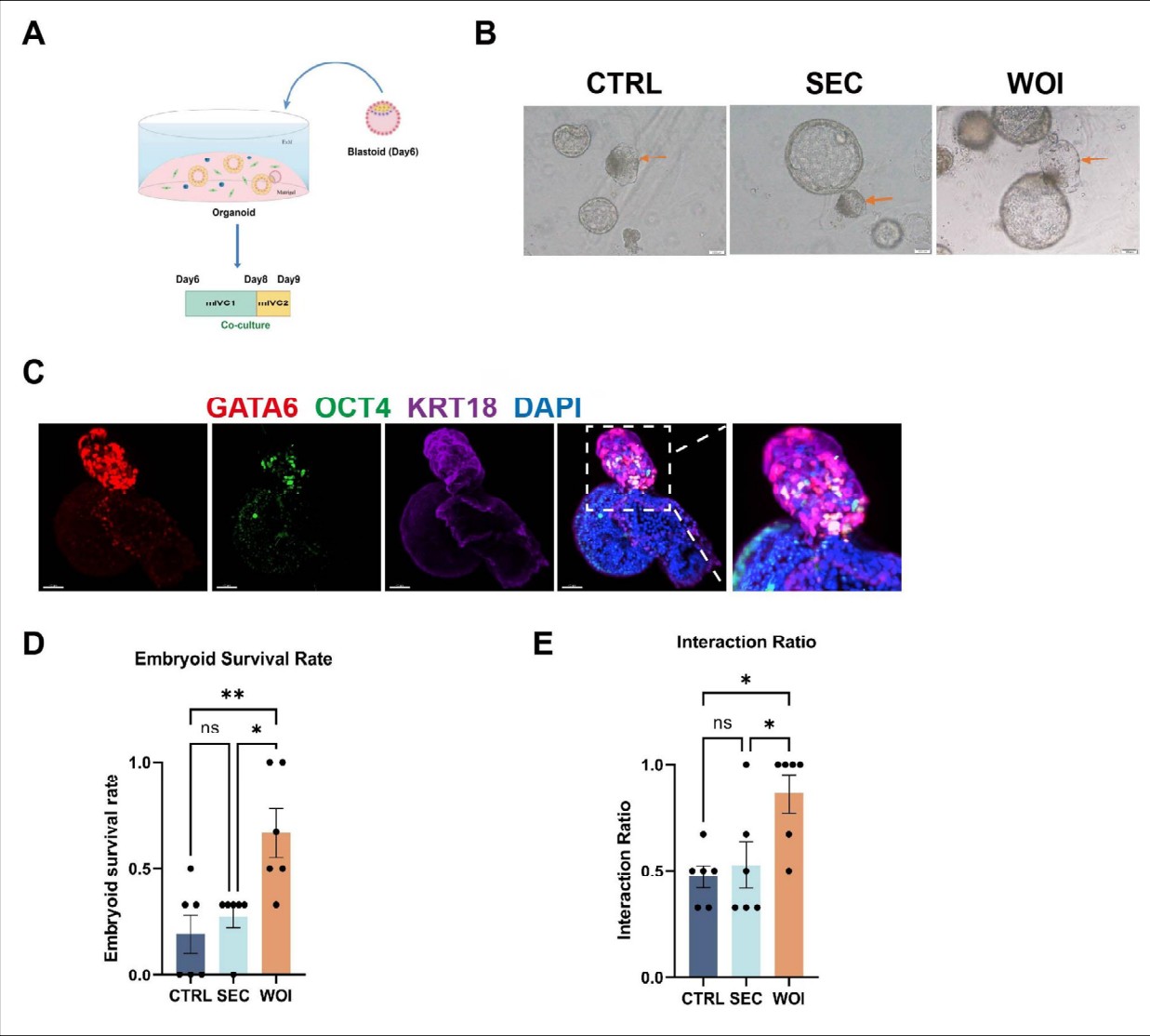

**Figure 6.** The receptive endometrial assembloids possess the potential for embryo implantation. (**A**) Diagram illustrated the co-culture model of endometrial assembloids with blastoids (the blastoid stage corresponds to a 6 day post-fertilization human embryo, referred to as Day 6 here). mIVC1: modified In Vitro Culture Medium 1, mIVC2: modified In Vitro Culture Medium 2. (**B**) Bright-field images of the co-culture of CTRL, secretory (SEC), and window of implantation (WOI) assembloids with blastoids (Day 9) (yellow arrows indicate the blastoids). Scale bar = 100 μm. (**C**) Whole-mount fluorescence staining of Day 9 co-cultured embryoid and assembloid. OCT4 indicates the epiblast, GATA6 indicates the hypoblast, and KRT18 indicates the trophoblast. Scale bar = 40 μm and 20 μm (the rightmost image). (**D**) Comparison of the survival rates of Day 9 embryoids in CTRL, SEC, and WOI assembloids. n=6. (**E**) Comparison of the interaction ratios between Day 9 embryoids and endometrial assembloids in the CTRL, SEC, and WOI groups. *$p<0.05$, **$p<0.005$. n=6.

In summary, WOI endometrial assembloids not only exhibited the typical structural and molecular features of the implantation window but also demonstrated significant potential for embryo implantation.

## Discussion

In this study, we constructed the WOI endometrial assembloids and observed the remarkable resemblance in structure and function to the in vivo endometrium. The assembloids consist of three primary types of endometrial cells, specifically epithelial, stromal, and immune cells, with epithelial cells assembling into glands surrounded by immune cells and stromal cells. This conservation of cellular composition and tissue architecture provides a foundational basis for mimicking the receptive endometrium.

While previous studies have induced secretory phase transition in endometrial assembloids using E2, P4, and cAMP, methodologies for accurately simulating the WOI or mid-secretory endometrium remain inadequate and require further refinement. Prior research has indicated that placental signals can enhance endometrial assembloid differentiation (*Turco et al., 2017*), with PRL, hCG, and HPL implicated in decidualization, implantation, immunoregulation, and angiogenesis (*Auriemma et al., 2020*; *d'Hauterive et al., 2022*; *Bazer et al., 2018*). Specifically, PRL, synthesized by the adenohypophysis, endometrium, and myometrium, plays a vital role in implantation, immunoregulation, and angiogenesis, with the secretory endometrium producing PRL in response to MPA and E2, leading to ciliated cell formation and stromal cell decidualization (*Turco et al., 2017*; *Auriemma et al., 2020*). HCG, secreted by trophoblasts in early pregnancy, modulates decidual cells (*d'Hauterive et al., 2022*) and improves endometrial thickness and receptivity (*Nwabuobi et al., 2017*). The introduction of hCG has been shown to upregulate key receptivity-associated factors, such as endocytosis proteins, hypoxia-inducible factor 1 (HIF1), chemokines, and glycodelin (*Bielfeld et al., 2019*). HPL contributes to the development and function of uterine glands (*Bazer et al., 2018*). Consequently, supplementary PRL, hCG, and HPL in our system augmented hormone responsiveness and receptivity, promoted cellular differentiation, and ultimately generated a model more representative of the receptive endometrium.

Notably, WOI assembloids exhibited the characteristic ultrastructures, such as cilia. Motile cilia are present in endometrial epithelium throughout the human menstrual cycle (*Wang et al., 2020b*). These hair-like organelles extend from the cell surface and beat rhythmically, facilitating cell and tissue movement while driving fluid transport across the epithelium (*Zhou and Roy, 2015*). During endometrial decidualization, the number and length of cilia increase, a process driven by the two primary regulatory factors for embryo implantation: estrogen and progesterone (*Haider et al., 2019*; *Li et al., 2023*; *Li et al., 2022c*). Throughout the menstrual cycle, ciliated and unciliated epithelia undergo mutual transformation from the secretory phase (or mid-secretory phase) to the menstrual phase, and then to the proliferative phase. Ciliated cell abundance peaks in the early-to-mid-secretory endometrium (*Wang et al., 2020b*), with the mid-secretory phase exhibiting a higher proportion of ciliated cells as opposed to the early and late-secretory phases (*Garcia-Alonso et al., 2021*). However, with aging, the expression of cilia-related genes in the endometrium is downregulated (*Devesa-Peiro et al., 2022*). Recurrent implantation failure (RIF) patients often present with endometrial ciliary defects, leading to impaired decidualization and repeated implantation failure (*Li et al., 2022c*). Thus, cilia play a crucial role in endometrial decidualization and embryo implantation. Consistently, the WOI assembloid exhibited increased ciliated epithelium, along with enhanced assembly and motility of cilia.

Furthermore, WOI assembloids demonstrated energy and lipid metabolism patterns resembling those of the in vivo receptive endometrium. Enhanced energy metabolism involves monocarboxylic acid metabolism and mitochondrial oxidative phosphorylation. Monocarboxylic acids, exemplified by lactate, pyruvate, and ketone bodies, are vital metabolites in most mammalian cells. During the implantation window, elevated lactate levels mobilize endometrial monocarboxylic acid metabolism and function as a pregnancy-related signal, stimulating secretion in the epithelium. Subsequently, ATP, the primary product of energy metabolism, induces neighboring epithelial cells to release IL8, promoting decidualization of stromal cells (*Gu et al., 2020*). Concordantly, our WOI assembloids indeed possessed larger and functionally active mitochondria and produced much more ATP and IL8 than CTRL and SEC assembloids. Lipid metabolism, responsible for energy storage, signal transduction, cell proliferation, apoptosis, and membrane trafficking, plays a crucial role in endometrial receptivity and implantation, although its precise mechanisms remain incompletely understood (*King et al., 2021*; *Vilella et al., 2013*; *Braga et al., 2019*). Thus, WOI assembloids possessed metabolic characteristics of in vivo implantation window.

However, our WOI endometrial assembloids also exhibit some limitations. It is undeniable that the assembloids cannot perfectly replicate the in vivo endometrium, which comprises functional and basal layers with a greater abundance of cell subtypes, under superior regulation by hypothalamic-pituitary-ovarian (HPO) axis. Specifically, stromal and immune cells are challenging to stably passage, and their proportion is lower than in the in vivo endometrium. While the in vivo peri-implantation period exhibits intricate gene expression dynamics driven by systemic regulation, our models only partially recapitulate these changes, primarily in mitochondria- and cilia-associated genes. Nevertheless, to some extent, these WOI assembloids possess receptivity characteristics and can be utilized

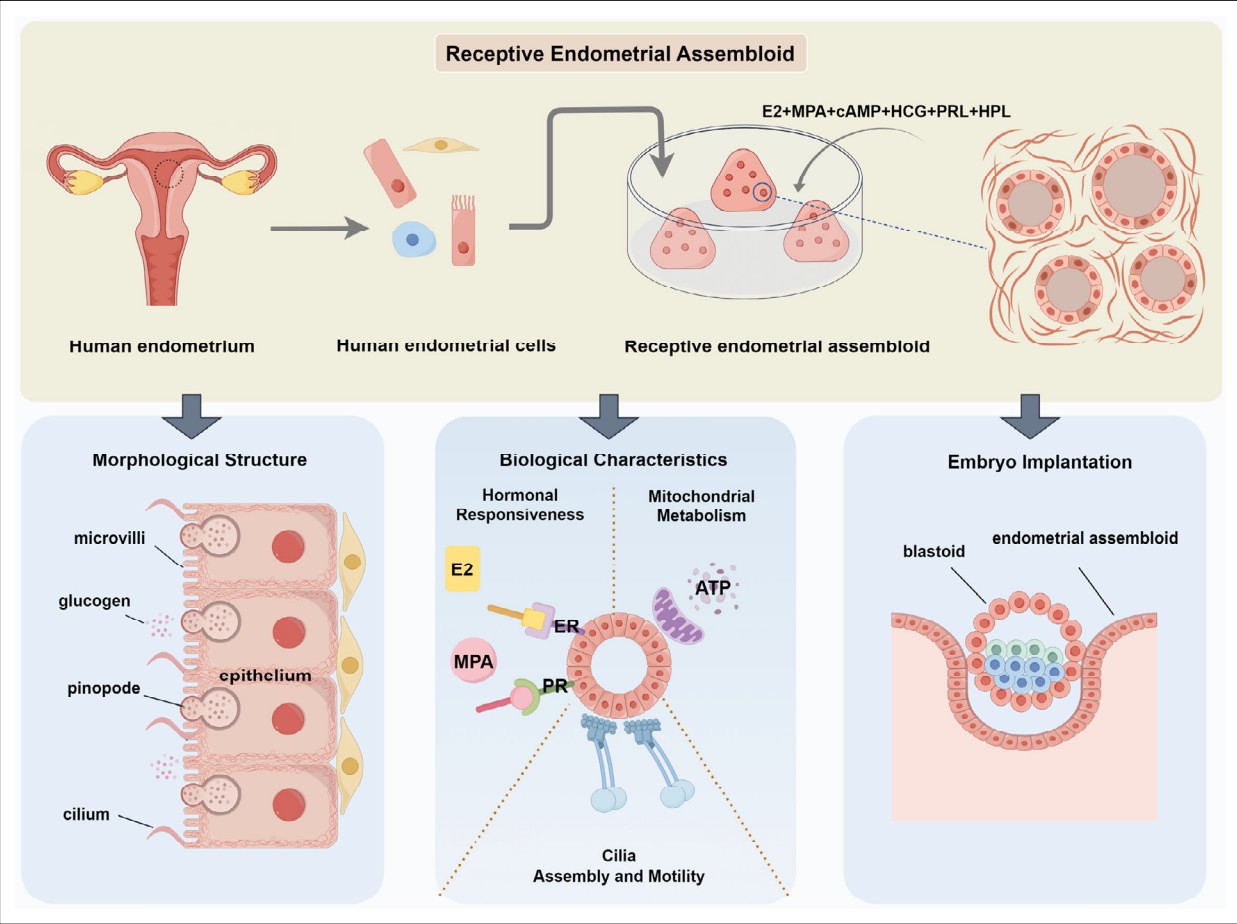

**Figure 7.** Schematic diagram displaying the establishment and validation of receptive endometrial assembloids, and summarizing the characteristic biological events of implantation-window endometrium.

for investigating receptivity-related scientific questions or conducting in vitro drug screening. Further refinements are required to fully simulate the dynamic endometrial gene expression patterns across all menstrual cycle stages. We are looking forward to integrating stem cell induction, 3D printing, and microfluidic systems to modify the culture environment.

In summary, we developed a receptive endometrial assembloid that mimics in vivo implantation-window endometrium (*Figure 7*). This model exhibits WOI-characteristic ultrastructures, including pinopodes and cilia, displays hormonal responsiveness and characteristic glycogen-secreting functions, recapitulates the processes, such as decidualization, metabolic changes and EMT, and retains embryo implantation potential. It serves as a valuable platform to investigate peri-implantation endometrial physiology and pathology, maternal-fetal interactions, with potential practical applications and clinical translation.

## Materials and methods
### Establishment of endometrial assembloids

All experiments involving human subjects followed medical ethical principles and the Declaration of Helsinki and were approved by the Institutional Review Board of Qilu Hospital of Shandong University (KYLL-202204–030). Informed consents were obtained from patients. Females of reproductive age who received hysterectomy for benign diseases were selected for this study. Clinical information of patients providing endometrial tissue was listed in *Source data 1*.

## Establishment of endometrial assembloids

Endometrium was aseptically collected into pre-cooled collection medium (DMEM/F12 (Gibco, 11039021)+10% FBS (Sigma, F0926)+1% antibiotic-antimycotic (Anti-Anti, Gibco, 15240–062)) stored temporarily at 4 °C. After washing with prechilled DPBS (Gibco, 14190136) and removing blood clots, the endometrium was minced in 1.5 ml EP tubes and transferred to digestion medium (DMEM/F12+1% antibiotic-antimycotic +0.4 mg/mL collagenase V (Sigma, C-9263)+1.25 U/mL dispase II (Sigma, D4693)+10 µg/ml DNase I (Worthington, LS002139)). Digestion occurred at 37 °C for 20 min, followed by neutralization with an equal volume of DMEM/F12 (10% FBS, 1% Anti-Anti). The digested suspension was vortexed, allowed to stand for 1 min, and filtered through a 40 µm cell strainer (Corning, 352340).

The cell strainer was inverted on the culture dish and rinsed with DPBS to collect the targeted cells. The resulting cell suspension was centrifuged at 400×g for 5 min. The supernatant was removed, 1 ml DMEM/F12 medium was added to resuspend the cell pellet, and the pellet was centrifuged again at 400×g for 5 min. After discarding the supernatant, the tube was chilled for 2–3 min, and cells were resuspended in DMEM/F12. Then, the appropriate amount of Matrigel (Corning, 536231) (volume ratio of cell suspension to Matrigel = 1:3) was added to the centrifuge tube and slowly pipetted to mix. Matrigel-cell compound was added slowly to the 24-well plate (1 drop per well, 40 µl/drop) and incubated at 37 °C for 30 min. Expansion medium (ExM) (*Supplementary file 1A*) (500 µL) was over-laid per well and changed every other day.

## Passage of endometrial assembloid

Assembloids were washed twice with DPBS after medium aspiration. A digestion solution was added, mixed with cells, and incubated at 37 °C for 20 min. An equal volume of neutralizing medium was added, and the cell suspension was centrifuged at 400×g for 5 min in a 15 ml tube. After discarding the supernatant, the cell sediments were washed with DMEM/F12 medium and centrifuged at 400 g for 5 min to acquire the sunken cells. The cell mass was resuspended in DMEM/F12, mixed with Matrigel (volume ratio of cell suspension to Matrigel = 1:3), and seeded into plates. ExM was added as previously described. The P1~P3 generation endometrial assembloids were used for downstream experiments.

## Cryopreservation of endometrial assembloids

Assembloids were cryopreserved during each passage. The assembloid culture was rinsed with DPBS, and then the assembloid was harvested with cell recovery solution (Corning 354253) after incubating on ice for less than 30 min. Following centrifugation at 400×g for 5 min, the assembloid was resuspended in serum-free, animal protein-free cell freezing medium (NCM Biotech, C40100), stored at –80 °C overnight, and transferred to liquid nitrogen for long-term storage.

To recover the assembloids, they were removed from liquid nitrogen, thawed at 37 °C, and resuspended in prewarmed DMEM/F12. The cell suspension was transferred to 15 ml tubes, diluted with additional medium, and centrifuged at 400×g for 5 min. Cell precipitates were washed with DMEM/F12, and cell-Matrigel suspensions were prepared and seeded into plates as previously described. Y-27632 (10 µM, Millipore scm075) was added to the assembloid medium for the first three cellular fluid exchanges and conventional medium thereafter.

## Hormone treatment of endometrial assembloids

Hormonal treatment was initiated following the assembly of the endometrial assembloids (about 7 day growth period). The hormone regimen for inducing endometrial secretory phase, as described by *Fitzgerald et al., 2019*, consisted of estradiol for 2 days followed by a combination of estradiol (Sigma E2758), medroxyprogesterone acetate (Selleck S2567) and N6,2'-O-dibutyryladenosine 3',5'-cyclic monophosphate sodium salt (cAMP) (Sigma D0627) for 6 days.

To simulate the endometrium of the implantation window, we developed a model by incorporating various pregnancy-related hormones. The hormone regimen included estradiol for the first two days, followed by a combination of estradiol, medroxyprogesterone acetate, cAMP, Human Chorionic Gonadotropin (HCG) (Livzon Pharmaceutical Group Inc), Human Placental Lactogen (HPL) (R&D Systems 5757-PL), and prolactin (Peprotech 100–07) for 6 days. The CTRL group was cultured in ExM

without hormone supplementation and subjected to parallel culture for 8 days along with the two aforementioned groups (*Figure 1A*; *Supplementary file 1B*).

## Long-term live-cell imaging and analysis

Endometrial assembloids from each group were placed in the Etaluma LS720 Microscope (Bio-Reach Co., Ltd.) for continuous live-cell imaging. The growth dynamics of the assembloids were monitored, and changes in assembloid counts, area, and average intensity were recorded over time to assess their developmental progression. The average intensity reflects the growth status of the assembloids by measuring their gray value. When assembloids undergo apoptosis, they typically condense into increasingly dense solid spheres, resulting in an elevated gray value (average intensity).

## Frozen section, periodic acid-Schiff staining, and immunofluorescence analysis

Assembloids were fixed in 4% PFA for 30 min, washed with PBS, dehydrated in 20% sucrose overnight at 4 °C, and embedded in OCT. The embedded assembloid was sectioned using a Thermo frozen microtome at a thickness of 10 μm. The sections were applied to periodic acid-Schiff staining (PAS) (Solarbio G1280).

As for immunofluorescence staining, sections were brought to room temperature, fixed with 4% PFA for 5 min, and permeabilized with 0.3% Triton X-100 in PBS for 20 min. Antigen retrieval was conducted using sodium citrate at 95 °C for 20 min. After being lowered to room temperature, the sections were blocked in QuickBlock Blocking Buffer for Immunol Staining (Beyotime Biotechnology, P0260) for 20 min. Sections were incubated with primary antibodies (Appendix 1—key resources table) at 4 °C overnight and then washed three times with PBS containing 0.1% Triton X-100. Secondary antibodies (Appendix 1—key resources table) were incubated at room temperature for 2 hr, followed by three washes with 0.1% Triton X-100 in PBS. The slices were incubated with DAPI (Beyotime Biotechnology C1002) for 15 min and mounted with an anti-fluorescence quencher. The pictures were collected by confocal laser scanning microscope (Andor Dragonfly 200) and processed with Imaris x64 9.0.1.

We quantified the fluorescence images using ImageJ. First, preprocess them by adjusting brightness and contrast and removing background noise with the 'Subtract Background' feature. Second, set the threshold to highlight the cells, then select the regions of interest (ROI) using selection tools. Third, as for counting the cells, navigate to Analyze > Analyze Particles. As for measuring the influence intensity and area, set the 'Measurement' options as mean gray value. Adjust parameters as needed, and view results in the 'Results' window. Save the data for further analysis and ensure consistency throughout your measurements for reliable results.

DAPI was used as an internal reference for normalization, where both DAPI and target fluorescence channel intensities were quantified simultaneously. The normalized target signal intensity (target/DAPI ratio) was then compared across experimental groups. A minimum of 15 images were analyzed for each parameter per group.

## Assembloid clearing and 3D imaging

The assembloids were fixed and cleared following Hans Clevers' Nature Protocols (*Dekkers et al., 2019*). Primary and secondary antibodies are listed in Appendix 1—key resources table, respectively. The images were captured by a light-sheet microscope (Carl Zeiss, Lightsheet 7) and analyzed via ZEN Blue software.

For 3D fluorescence quantification, ZEN software (Carl Zeiss) was exclusively used, with 11 images analyzed per experimental group.

## Bulk transcriptome sequencing of endometrial assembloids

One assembloid was removed from the droplet, washed several times with DPBS, and placed in lysis buffer using a capillary glass pipette under a dissecting microscope. The synthesis and amplification of full-length cDNAs were performed following the Smart-seq2 protocol (*Picelli et al., 2014*). The total RNA of the assembloid was used as the template for the first cDNA synthesis, and Oligo dT Primer was used as the reverse transcription primer. A splice sequence was added to the 3' end of cDNA using template-switching activity of Discover-sc Reverse Transcriptase. The RT reaction was normally

performed at 42 °C for 90 min and 70 °C for 15 min. After the first-strand reaction, the cDNA was amplified using nine cycles and purified. The concentration of each library was measured by Qubit. Check the size distribution on an Agilent high-sensitivity chip. Sequencing was performed on an Illumina X-ten platform (Gene Denovo, Guangzhou, China).

## Micro proteomics of endometrial assembloids

Assembloids were digested, collected, and washed with DPBS. At least $1 \times 10^5$ cells were transferred to low-attachment PCR tubes (Axygen, PCR-02-L-C). Downstream 4D label-free proteomic quantification was conducted by Jingjie PTM Biolab Co., Inc (HangZhou, China). Each sample was added lysis solution, non-contact sonicated for 3 min and heated at 95 °C for 10 min. When restored to room temperature, the sample was digested in 10 ng/µL trypsin overnight at 37 °C. The protein solution was reduced with 5 mM dithiothreitol for 30 min at 56 °C and alkylated with 11 mM iodoacetamide for 15 min at room temperature in darkness. Then the protein was analyzed by a combination of liquid chromatography and mass spectrometry. The tryptic peptides were dissolved in solvent A (0.1% formic acid, 2% acetonitrile/in water), directly loaded onto a home-made reversed-phase analytical column (25 cm length, 75/100 µm i.d.). Peptides were separated with a gradient from 6% to 24% solvent B (0.1% formic acid in acetonitrile) over 70 min, 24 to 35% in 12 min and climbing to 80% in 4 min then holding at 80% for the last 4 min, all at a constant flow rate of 450 nL/min on a nanoElute UHPLC system (Bruker Daltonics). The peptides were subjected to a capillary source followed by the timsTOF Pro (Bruker Daltonics) mass spectrometry. The electrospray voltage applied was 1.75 kV. Precursors and fragments were analyzed at the TOF detector, with a MS/MS scan range from 100 to 1700 m/z. The timsTOF Pro was operated in parallel accumulation serial fragmentation (PASEF) mode. Precursors with charge states 0–5 were selected for fragmentation, and 10 PASEF-MS/MS scans were acquired per cycle. The dynamic exclusion was set to 30 s.

## Endometrial receptivity analysis test of endometrial assembloids

Endometrial assembloids were obtained from Matrigel via cell recovery solution and placed into RNAlater buffer (AM7020; Thermo Fisher Scientific, Waltham, MA, USA). Total RNA was extracted using the RNeasy Micro Kit (Qiagen 74004). Quantification and quality were assessed by a Qubit High Sensitivity RNA Kit (Thermo Fisher Scientific Q32855) and Agilent 2100 Bioanalyzer. Assembloids with RNA integrity number (RIN)>7 were used for downstream experiments. RNA reverse transcription, cDNA library construction, and endometrial receptivity determination were performed by Yikon Genomics (Jiangsu, China) (*He et al., 2021*). The MALBAC Platinum Single Cell RNA Amplification Kit (KT110700796; Yikon Genomics, Suzhou, Jiangsu, China) was used for RNA reverse transcription. cDNA lengths of 1000–10,000 bp were in accordance with the quality control requirements. Library construction was performed using the Gene Sequencing and Library Preparation Kit (XY045; Yikon Genomics, Suzhou, Jiangsu, China). Single-end sequencing was performed on a HiSeq 2500 platform (Illumina, San Diego, CA, USA). The read length was set to 140 bp. The capacity of the raw data was approximately 5 M reads. The DEGs of every sample were put into the predictive model to predict the endometrial receptivity. ERA was employed specifically as a confirmatory test for the WOI assembloids, rather than as a comparative measure across all groups.

## Single-cell transcriptome sequencing and analysis of endometrial assembloids

The CTRL, SEC, and WOI endometrial assembloids from one patient were digested with digestion medium similar to digestion during passaging, but the duration of digestion was extended to 30 min. The cell precipitate obtained by centrifugation continued to be digested with 0.25% trypsin (HyClone, SH30042.01) into single cell suspensions. An equal volume of neutralizing medium was added to halt the digestion and centrifuged to acquire the cells. Cells were resuspended in PBS containing 0.1% BSA, and the cellular suspension was passed through a 40 µm cell strainer. The filtered cells were collected and subjected to cell counting and viability assays.

Cellular suspensions were loaded on a 10 X Genomics GemCode Single-cell instrument that generates single-cell Gel Bead-In-Emulsion (GEMs). The resulting GEMs were processed by Gene Denovo Biotechnology Co. (Guangzhou, China). Libraries were generated and sequenced from the cDNAs with Chromium Next GEM Single Cell 3' Reagent Kits v3.1. The cellular gene matrices of each sample,

produced via unique molecular identifier (UMI) counting and cell barcodes, were individually imported into Seurat version 3.1.1 for downstream analysis. Cells with an unusually high number of UMIs (≥8000) or mitochondrial gene percentage (≥10%) were filtered out. We also excluded cells with fewer than 500 or more than 4000 genes detected. Ultimately, we acquired 8608, 8932, and 9243 cells in the CTRL, SEC, and WOI groups, respectively, with more than 40,000 reads and approximately 2000 genes per cell after quality control (*Figure 2—figure supplement 1A*). The single-cell trajectory was analyzed using a matrix of cells and gene expression by Monocle (Version 2.10.1) (*Trapnell et al., 2014*). We used CellphoneDB to analyze the expression abundance of ligand-receptor interactions between two cell types on the basis of the expression of a receptor by one cell type and a ligand by another cell type (*Efremova et al., 2020*). The human transcription factor database (TFDB) was applied for annotating transcription factors to explore downstream gene expression and upstream epigenetic regulation in cells.

## Single-cell transcriptome analysis combined with published dataset

The data quality of four raw datasets (CTRL, SEC, and WOI assembloids, and mid-secretory endometrium) were assessed using Cellranger, followed by a comparison with the reference genome of *Homo sapiens* Ensembl_release103 to derive the expression matrix. Next, the expression matrices of samples were imported into Seurat. Cells were filtered using various metrics to ensure the retention of high-quality cells. This included the removal of doublets using DoubletFinder and the elimination of aberrant cells based on gene expression, UMI count, and mitochondrial gene ratio. Subsequently, we applied Harmony for batch effect correction and data integration. The dataset underwent dimensionality reduction through PCA. The soft k-means clustering algorithm was employed to address batch effects and clustering, utilizing a clustering parameter resolution of 0.5. Finally, the clustering results were visualized using t-SNE or UMAP based on the cell subpopulation classification.

## Flow cytometric analysis

Flow cytometry was performed to investigate the distribution of T cells and macrophages in the endometrial assembloid using post-digestive single-cell suspensions. T cells and macrophages were then stained using different staining panels (Appendix 1—key resources table).

For staining of T cells, $7 \times 10^4$ single cells were washed by ice-cold staining buffer (PBS containing 2% fetal bovine serum), and stained for surface markers with antibodies, including PerCP-Cy 5.5-conjugated anti-human CD45 antibody, V450-conjugated anti-human CD3 antibody, BV510-conjugated anti-human CD4 antibody, and APC/Cyanine7-conjugated anti-human CD8 antibody (Appendix 1—key resources table) for 30 min at 4 °C in the dark, following incubation with Human TruStain FcX (Fc Receptor Blocking Solution) (Biolegend 422301) for 10 min at room temperature. Cells were then stained with Zombie Green Fixable Viability Kit (Biolegend 423111) in the dark for 20 min at room temperature.

For staining of macrophages, $7 \times 10^4$ single cells were dealt with as mentioned above except for staining for surface markers with PerCP-Cy 5.5-conjugated anti-human CD45 antibody, Brilliant Violet 785-conjugated anti-human CD68 antibody, and Brilliant Violet 650-conjugated anti-human CD11b antibody (Appendix 1—key resources table) and staining with Zombie NIR Fixable Viability Kit (Biolegend 423105).

After staining, cells were fixed by 4% paraformaldehyde (PFA), washed once and then resuspended in 300 µl PBS followed by flow cytometric analysis on a BD LSR Fortessa instrument (BD Bioscience). Data were all analyzed by FlowJoV10 (BD Bioscience) in this study.

## Transmission electron microscopy

Samples were rinsed quickly with PB buffer, immediately placed into 3% glutaraldehyde fixation solution (pH 7.4), and trimmed to 1 mm×1 mm×3 mm. The following experiments were carried out at Weiya Electron Microscopy Laboratory (Jinan, Shandong, China). Rinse sequentially according to the standard TEM sample preparation methods. Then the samples were postfixed in 1% osmium tetroxide, dehydrated in a series of acetone, infiltrated, and embedded with Epon 812. After semi-thin sectioning, use an ultramicrotome (LKB-V, LKB Company, Sweden) for 70–100 nm ultrathin sectioning. The sections were stained with lead citrate and uranyl acetate and observed using a transmission

electron microscope (JEM-1200EX; JEOL Ltd., Tokyo, Japan). The images were recorded using a CCD camera (MORADA-G2, Olympus Corporation, Japan).

Pinopodes are large, bulbous protrusions with a smooth apical membrane. Under transmission electron microscopy (TEM), it can be observed that the pinopodes contain various small particles, which are typically extracellular fluid and dissolved substances.

Microvilli are elongated, finger-like projections that typically exhibit a uniform and orderly arrangement, forming a 'brush border' structure. Under transmission electron microscopy, dense components of the cytoskeleton, such as microfilaments and microtubules, can be seen at the base of the microvilli.

The cilium is composed of microtubules. The cross-section shows that the periphery of the cilium is surrounded by nine pairs of microtubules arranged in a ring. The longitudinal section shows that the cilium has a long cylindrical structure, with the two central microtubules being quite prominent and located at the center of the cilium.

For TEM-observed pinopodes, glycogen particles, microvilli, and cilia, manual region-of-interest (ROI) selection was performed using ImageJ software for quantitative analysis of counts, area, and length. Twenty randomly selected images per experimental group were analyzed for each morphological parameter.

## Real-time quantitative PCR

Total RNA was extracted and its concentration was determined as above bulk transcriptome sequencing methods. Genomic DNA contamination was eliminated using the gDNA wiper Mix (Vazyme, R323). Total RNA (500 ng) from each sample was transcribed in a total reaction volume of 20 µL using HiScript III qRT SuperMix (Vazyme, R323). Real-time PCR was performed using SYBR Green Premix Pro Taq HS qPCR Kit (Accurate Biotechnology (human) Co., Ltd AG11701) and PCR primers (BioSune) (Appendix 1—key resources table) in a 10 µL reaction volume. GAPDH was used as a reference gene.

## Proximity ligation assay (PLA)

The frozen sections were pretreated for fixation, permeabilized, and antigen retrieval as above immunofluorescence staining. The following steps were carried out according to the Duolink PLA Fluorescence Protocol (Sigma DUO92008, DUO82049).

The sample was blocked with Blocking Solution and incubated at 37°C for 1 hr. Discard the Blocking Solution and incubate with the primary antibody (Appendix 1—key resources table) at 4°C overnight. On the second day, remove the primary antibody, clean with Wash Buffer A, and incubate with PLUS and MINUS PLA probe (PLUS: MINUS PLA probe = 1:5) at 37 °C for 1 hr. The sample was then incubated with Ligation Solution at 37 °C for 30 min, followed by amplification with Amplification Solution at 37 °C for 100 min. After being washed with Wash Buffer B, the slice was sealed by Duolink PLA Mounting Medium with DAPI. The images were captured under the confocal laser scanning microscope (Andor Dragonfly 200) and processed with Imaris x64 9.0.1.

## ATP detection

Assembloids were digested (as described above) to obtain cell pellet. The following steps were carried out according to the ATP Test Kit (Beyotime S0026). The lysate was added to lyse the cell pellet (200 µl/$2 \times 10^5$ cells), followed by centrifugation at 4°C 12,000 g for 5 min. The resulting supernatant was used for subsequent detection.

Preparation of the standard curve: ATP standard solution of 0.01, 0.03, 0.1, 0.3, 1, 3, and 10 µM can be detected to acquire the standard curve.

Preparation of ATP test working solution: Take an appropriate amount of ATP test reagent and dilute it with ATP test reagent diluent at the ratio of 1:9.

Determination of ATP concentration: Add 100 µl of ATP test solution to the test hole at room temperature for 3–5 min to reduce the background. Add 20 µl of sample or standard to the test hole and quickly mix them, which was followed by detecting the relative light units (RLU) value with the luminometer. The concentration of ATP in the sample was calculated according to the standard curve.

## Human IL-8 detection

The medium of each well assembloids was collected and centrifuged at 1000 g for 10 min. The supernatant was put into conducted follow-up experiments according to the instructions of Human IL-8/CXCL8 ELISA Kit (ABclonal RK00011).

Wash the microwell, add 100 μL standard/sample diluent (R1) into the blank well, then add 100 μL standard or sample with different concentrations into the other holes. The microwell plate was incubated at 37°C for 2 hr. Discard the liquid in the well and wash the microwell. Biotinized antibody working solution (100 μL/ well) was added to each well and incubated at 37°C for 1 hr. Discard the liquid in the well and wash the microwell. Streptavidin-HRP working solution (100 μL/ well) was added to each well and incubated at 37°C for 30 min. Discard the liquid in the well and wash the microwell. Add TMB substrate (100 μL/well) to the microwell, which was incubated at 37 °C for 15–20 min in the dark. Add the termination solution (50 μL/well), then immediately put the plate into the microplate reader and determine the OD value of 450 nm within 5 min. The correction wavelength was set to 570 nm. Subtract the 570 nm reading from the 450 nm reading to correct and remove the OD value of the non-chromogenic substance, resulting in a more accurate detection result. The concentration of IL8 for each sample is calculated according to the standard curve.

## Co-culture of blastoids and endometrial assembloids

Blastoids were generated following the method reported in the 2021 Nature (*Yu et al., 2021*). The endometrial assembloids culture medium was aspirated, and pre-equilibrated modified In Vitro Culture Medium 1 (mIVC1) was added (*Supplementary file 1C*). On a 37 °C warming plate, Day 6 blastoids were placed into endometrial assembloids using a glass needle and immediately transferred to an incubator maintained at 37 °C, 6% $CO_2$, and saturated humidity. Subsequently, half-medium changes were performed daily. At Day 8 of culture, the medium was switched to mIVC2 (*Supplementary file 1C*), and culture was continued until Day 9. Blastoids that remained firmly attached to endometrial assembloids during medium changes and fluorescence staining were considered to have undergone interaction with assembloids. Interaction rate = (Number of interacting embryoid-like structures/Total number of embryoid-like structures)×100%.

All biological replicates (fourteen individuals) of endometrial assembloids show similar expression of receptivity markers and comparable capacity to support blastoid attachment.

## Statistical analysis

Statistical analysis was conducted with GraphPad Prism 9 (GraphPad Software Inc, La Jolla, CA, USA). Multiple group comparisons were analyzed by one-way ANOVA and subsequent Tukey's multiple comparison test. Data are expressed as mean ± SEM, and $P < 0.05$ was defined as statistically significant.

## Acknowledgements

This work was supported by the National Natural Science Foundation of China (82192874 and 82502017), the National Key R&D Program of China (2023YFA1801803), the Excellence Research Group Program of NSFC (32588201), the Key R&D Program of Shandong Province, China (2024CXPT081), CAMS Innovation Fund for Medical Sciences (2021-I2M-5-001), the Fundamental Research Funds of Shandong University (2023QNTD004), the Fundamental Research Funds of Shandong University Taishan Scholars Program of Shandong Province (ts20190988 and tsqn201909194). We sincerely appreciate Prof. Tianqing Li (Kunming University of Science and Technology) and Prof. Shaorong Gao (Tongji University) for insightful comments and discussions. We are grateful to Guangzhou Genedenovo Biotechnology Co., Ltd. for assisting in sequencing and bioinformatics analysis. We also thank YiKon Medical from China for technological assistance of ERA, Jingjie PTM Biolab (Hangzhou) Co., Inc for proteomics.

# Additional information

## Funding

| Funder | Grant reference number | Author |
|---|---|---|
| National Natural Science Foundation of China | 82192874 | Han Zhao |
| National Key Research and Development Program of China | 2023YFA1801803 | Keliang Wu |
| Excellence Research Group Program of NSFC | 32588201 | Zi-Jiang Chen |
| Key Research and Development Program of Shandong Province | 2024CXPT081 | Zi-Jiang Chen |
| Fundamental Research funds of Shandong University Taishan Scholars Program of Shandong Province | ts20190988 | Han Zhao |
| Fundamental Research funds of Shandong University Taishan Scholars Program of Shandong Province | tsqn201909194 | Keliang Wu |
| CAMS Innovation Fund for Medical Sciences | 2021-I2M-5-001 | Zi-Jiang Chen |
| National Natural Science Foundation of China | 82502017 | Yu Zhang |
| Fundamental Research Fund of Shandong University | 2023QNTD004 | Keliang Wu |

The funders had no role in study design, data collection and interpretation, or the decision to submit the work for publication.

## Author contributions

Yu Zhang, Conceptualization, Data curation, Formal analysis, Validation, Writing - original draft, Writing – review and editing; Rusong Zhao, Conceptualization, Supervision, Writing – review and editing; Chaoyan Yang, Jinzhu Song, Changjian Yin, Zhenzhen Hou, Chuanxin Zhang, Methodology; Peishu Liu, Tao Li, Resources; Yan Li, Boyang Liu, Writing – review and editing; Minghui Lu, Validation; Zi-Jiang Chen, Keliang Wu, Conceptualization, Funding acquisition; Han Zhao, Conceptualization, Funding acquisition, Project administration, Writing – review and editing

## Author ORCIDs

Yu Zhang ⓘ https://orcid.org/0000-0003-0268-0137
Yan Li ⓘ https://orcid.org/0000-0003-1813-8115
Boyang Liu ⓘ https://orcid.org/0000-0002-1285-4842
Zi-Jiang Chen ⓘ https://orcid.org/0000-0001-6637-6631
Han Zhao ⓘ https://orcid.org/0000-0001-9515-7534

## Ethics

All experiments involving human subjects followed medical ethical principles and the Declaration of Helsinki and were approved by the Institutional Review Board of Qilu Hospital of Shandong University (KYLL-202204-030). Informed consents were obtained from patients.

Reviewer #2 (Public review): https://doi.org/10.7554/eLife.90729.5.sa1
Author response https://doi.org/10.7554/eLife.90729.5.sa2

# Additional files

## Supplementary files

Supplementary file 1. Composition of medium. Table A Composition of expansion medium (ExM) of endometrial assembloid. Table B Composition of hormone regimen of endometrial assembloid. Table C Composition of modified In Vitro Culture Medium (mIVC1 and mIVC2) for co-culture of blastoids and endometrial assembloids.

MDAR checklist

Source data 1. Clinical information of patients and lists of key genes and proteins for focused analysis.

## Data availability

RNA sequence data were deposited in National Genomics Data Center of China (accession number HRA007224 and HRA007214). Requests for access will be reviewed by the Data Access Committee (DAC). All data generated or analyzed during this study are included in the manuscript and supporting files; source data files have been provided for all figures. Single-cell transcriptomic sequencing data of endometrium with reference datasets GSE111976 and E-MTAB-10287.

The following datasets were generated:

| Author(s) | Year | Dataset title | Dataset URL | Database and Identifier |
| --- | --- | --- | --- | --- |
| Zhang Y, Zhao R, Yang C, Song J, Liu P, Li Y, Liu B, Li T, Yin C, Lu M, Hou Z, Zhang C, Chen Z-J, Wu K, Zhao H | 2026 | Human receptive endometrial organoid for deciphering the implantation window | https://ngdc.cncb. ac.cn/gsa-human/ browse/HRA007214 | National Genomics Data Center of China, HRA007214 |
| Zhang Y, Zhao R, Yang C, Song J, Liu P, Liu B, Li Y, Li T, Yin C, Lu M, Hou Z, Zhang C, Chen Z-J, Wu K, Zhao H | 2026 | Human receptive endometrial organoid for deciphering the implantation window- scRNAseq | https://ngdc.cncb. ac.cn/gsa-human/ browse/HRA007224 | National Genomics Data Center of China, HRA007224 |

The following previously published datasets were used:

| Author(s) | Year | Dataset title | Dataset URL | Database and Identifier |
| --- | --- | --- | --- | --- |
| Wang W, Vilella F, Alama P, Moreno I, Isakova A, Pan W, Simon C, Quake SR | 2020 | Single cell RNA-seq analysis on human endometrium across the natural menstrual cycle | https://www.ncbi. nlm.nih.gov/geo/ query/acc.cgi?acc= GSE111976 | NCBI Gene Expression Omnibus, GSE111976 |

*Continued on next page*

*Continued*

| Author(s) | Year | Dataset title | Dataset URL | Database and Identifier |
|---|---|---|---|---|
| Garcia-Alonso L, Handfield L-F, Nikolakopoulou K, Arutyunyan A, Li T, Kleshchevnikov V, Kwakwa K, Tuck E, Roberts K, Fernando RC, Polanski K, Hoo R, Tarkowska A, Lorenzi V, Porter T, Gardner L, Sancho-Serra C, Massalha H, Woodhams B, Mazzeo CI, Prete M, van Dongen S, Dabrowska M, Vaskivskyi V, Mahbubani KT, Park J, Jimenez-Linan M, Ayuk P, Moore L, Bayraktar OA, Teichmann SA, Vento-Tormo R | 2021 | Single-cell RNA-seq of Endometrial Superficial biopsies and full depth uterine tissue from organ donors | https://www.ebi.ac.uk/biostudies/arrayexpress/studies/E-MTAB-10287 | Arrayexpress, E-MTAB-10287 |

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

# Appendix 1

## Appendix 1—key resources table

| Reagent type (species) or resource | Designation | Source or reference | Identifiers | Additional information |
|---|---|---|---|---|
| Biological sample (*Homo sapiens*) | Primary human endometrium | This paper | - | See Materials and methods |
| Antibody | E-Cadherin (4A2) (Mouse monoclonal) | Cell Signaling | Cat#:14472 RRID:AB_2728770 | IF (1:200) |
| Antibody | Vimentin (D21H3) XP (Rabbit monoclonal) | Cell Signaling | Cat#: 5741 S RRID:AB_10695459 | IF (1:200) |
| Antibody | Ki-67 (D3B5) (Rabbit monoclonal) | Cell Signaling | Cat#: 9129 RRID:AB_2687446 | IF (1:400) |
| Antibody | Cleaved Caspase-3 (Asp175) Antibody (Rabbit monoclonal) | Cell Signaling | Cat#: 9661 S RRID:AB_2341188 | IF (1:400) |
| Antibody | Progesterone Receptor A/B (D8Q2J) XP (Rabbit monoclonal) | Cell Signaling | Cat#: 8757 S RRID:AB_2797144 | IF (1:800) |
| Antibody | FoxO1 (D7C1H) (Mouse monoclonal) | Cell Signaling | Cat#:14952 S RRID:AB_2722487 | IF (1:100) |
| Antibody | Rabbit anti-FOXA2 (Rabbit monoclonal) | Abcam | Cat#: ab108422 RRID:AB_11157157 | IF (1:500) |
| Antibody | Anti-PDGF (Rabbit polyclonal) | Abcam | Cat#: ab23914 RRID:AB_2162180 | IF (1:160) |
| Antibody | Anti-Integrin alpha 5 (Rabbit monoclonal) | Abcam | Cat#: ab150361 RRID:AB_2631309 | IF (1:250) |
| Antibody | Anti-CD44 (Mouse monoclonal) | Abcam | Cat#: ab254530 RRID:AB_2885131 | IF (1:500) |
| Antibody | Anti-IGFBP1 (Rabbit polyclonal) | Abcam | Cat#: ab228741 RRID:AB_3665797 | IF (1:400) |
| Antibody | Mouse anti-Estrogen Receptor alpha (Mouse monoclonal) | Santa Cruz | Cat#: sc-8002 RRID:AB_627558 | IF (1:50) |
| Antibody | Neuropilin-2 antibody (Mouse monoclonal) | Santa Cruz | Cat#: sc-13117 RRID:AB_628044 | IF (1:50) |
| Antibody | TPPP antibody (Mouse monoclonal) | Santa Cruz | Cat#: sc-515819 RRID:AB_10611323 | IF (1:50) |
| Antibody | MAO-A (Mouse monoclonal) | Santa Cruz | Cat#: sc-271123 RRID:AB_10609510 | IF (1:50) |
| Antibody | Goat anti-Human DPPIV/CD26 (Goat polyclonal) | R&D Systems | Cat#: AF1180 RRID:AB_354651 | IF (1:100) |
| Antibody | HIF-1 alpha antibody (Mouse monoclonal) | NOVUS | Cat#: NB 100–105 RRID:AB_350048 | IF (1:50) |
| Antibody | Acetyl-α-tubulin (Mouse monoclonal) | Sigma | Cat#: T7451 RRID:AB_609894 | IF (1:20000) |
| Antibody | Anti-ODF2 antibody (Rabbit polyclonal) | Sigma | Cat#: HPA 048841 RRID:AB_1079522 | IF (1:100) |
| Antibody | SLC25A1 Polyclonal antibody (Rabbit polyclonal) | Proteintech | Cat#: 15235–1-AP RRID:AB_2254794 | IF (1:500) |
| Antibody | CD45 Monoclonal Antibody (30-F11) (Mouse monoclonal) | eBioscience | Cat#: 14-0451-81 RRID:AB_467250 | IF (1:100) |
| Antibody | Donkey anti-Mouse IgG (H+L) Highly Cross-Adsorbed Secondary Antibody Alexa Fluor Plus 488 | Thermo | Cat#: A32766 RRID:AB_2762823 | IF (1:1000) |
| Antibody | Donkey anti-Mouse IgG (H+L) Highly Cross-Adsorbed Secondary Antibody Alexa Fluor Plus 555 | Thermo | Cat#: A32773 RRID:AB_2762848 | IF (1:10000) |

*Appendix 1 Continued on next page*

*Appendix 1 Continued*

| Reagent type (species) or resource | Designation | Source or reference | Identifiers | Additional information |
|---|---|---|---|---|
| Antibody | Donkey anti-Rabbit IgG (H+L) Highly Cross-Adsorbed Secondary Antibody Alexa Fluor Plus 555 | Thermo | Cat#: A32794 RRID:AB_2762834 | IF (1:1000) |
| Antibody | Donkey anti-Rabbit IgG (H+L) Highly Cross-Adsorbed Secondary Antibody Alexa Fluor Plus 488 | Thermo | Cat#: A32790 RRID:AB_2762833 | IF (1:1000) |
| Antibody | Donkey anti-Rabbit IgG (H+L) Highly Cross-Adsorbed Secondary Antibody Alexa Fluor Plus 647 | Thermo | Cat#: A32795 RRID:AB_2762835 | IF (1:1000) |
| Antibody | Donkey anti-Goat IgG (H+L) Highly Cross-Adsorbed Secondary Antibody Alexa Fluor Plus 647 | Thermo | Cat#: A32849 RRID:AB_2762840 | IF (1:1000) |
| Antibody | Goat anti-Rat IgG (H+L) Cross-Adsorbed Secondary Antibody, Alexa Fluor 488 | Thermo | Cat#: A11006 RRID:AB_2534074 | IF (1:500) |
| Antibody | BD Pharmingen PerCP-Cy5.5 Mouse Anti-Human CD45 (Mouse monoclonal) | BD Bioscience | Cat#: 564105 RRID:AB_2744405 | FACS (1:200) |
| Antibody | BD Horizon V450 Mouse Anti-Human CD3 (Mouse monoclonal) | BD Bioscience | Cat#: 560365 RRID:AB_1645570 | FACS (1:200) |
| Antibody | BD Horizon BV510 Mouse Anti-Human CD4 (Mouse monoclonal) | BD Bioscience | Cat#: 562970 RRID:AB_2744424 | FACS (1:200) |
| Antibody | APC/Cyanine7 anti-human CD8 Antibody (Mouse monoclonal) | Biolegend | Cat#: 344714 RRID:AB_2044006 | FACS (1:200) |
| Antibody | CD127 Monoclonal Antibody (eBioRDR5), Alexa Fluor 700 (Mouse monoclonal) | invitrogen | Cat#: 56-1278-42 RRID:AB_2637327 | FACS (1:200) |
| Antibody | Brilliant Violet 785 anti-human CD68 Antibody (Mouse monoclonal) | Biolegend | Cat#: 333825 RRID:AB_2800879 | FACS (1:200) |
| Antibody | Brilliant Violet 650 anti-human CD11b Antibody (Mouse monoclonal) | Biolegend | Cat#: 301335 RRID:AB_2562761 | FACS (1:200) |
| Antibody | SEMA3A antibody (C-1) (Mouse monoclonal) | Santa Cruz | Cat#: sc-74555 RRID:AB_2185394 | PLA (1:50) |
| Antibody | Recombinant Anti-Neuropilin 1 antibody [EPR3113] (Rabbit monoclonal) | Abcam | Cat#: ab81321 RRID:AB_1640739 | PLA (1:250) |
| Antibody | Recombinant Anti-ROR2 antibody [EPR19980] (Rabbit monoclonal) | Abcam | Cat#: ab218105 RRID not registered | PLA (1:500) |
| Antibody | Wnt-5a antibody (A-5) (Mouse monoclonal) | Santa Cruz | Cat#: sc-365370 RRID:AB_10846090 | PLA (1:50) |
| Antibody | Monoclonal Anti-CD74 antibody produced in mouse (Mouse monoclonal) | Sigma | Cat#: SAB5201932 RRID not registered | PLA (1:100) |
| Antibody | Recombinant Anti-alpha COP I/COPA Antibody [EPR14273(B)] (Rabbit monoclonal) | Abcam | Cat#: ab181224 RRID:AB_2893196 | PLA (1:50) |
| Antibody | CD46 antibody (M177) (Mouse monoclonal) | Santa Cruz | Cat#: sc-52647 RRID:AB_629109 | PLA (1:50) |
| Antibody | Jagged1 (28H8) Rabbit mAb (Rabbit monoclonal) | Cell Signaling | Cat#: 2620T RRID:AB_10693295 | PLA (1:1000) |
| Sequence-based reagent | PGR_F | This paper | PCR primers | ACCCGCCCTATCTCAACTACC |
| Sequence-based reagent | PGR_R | This paper | PCR primers | AGGACACCATAATGACAGCCT |
| Sequence-based reagent | PAEP_F | This paper | PCR primers | GAGATCGTTCTGCACAGATGG |
| Sequence-based reagent | PAEP_R | This paper | PCR primers | CGTTCGCCACCGTATAGTTGAT |

*Appendix 1 Continued on next page*

*Appendix 1 Continued*

| Reagent type (species) or resource | Designation | Source or reference | Identifiers | Additional information |
|---|---|---|---|---|
| Sequence-based reagent | OLFM4_F | This paper | PCR primers | ACCTTTCCCGTGGACA GAGT |
| Sequence-based reagent | OLFM4_R | This paper | PCR primers | TGGACATATTCCCTCACTTT GGA |
| Sequence-based reagent | ESR1_F | This paper | PCR primers | CCCACTCAACAGCGTG TCTC |
| Sequence-based reagent | ESR1_R | This paper | PCR primers | CGTCGATTATCTGAATTTGG CCT |
| Peptide, recombinant protein | Noggin | Proteintech | Cat. #: HZ-1118 | 100 ng/ml |
| Peptide, recombinant protein | EGF | Peprotech | Cat. #: AF-100–15 | 50 ng/ml |
| Peptide, recombinant protein | FGF2 | Origene | Cat. #: TP750002 | 100 ng/ml |
| Peptide, recombinant protein | WNT-3A | Proteintech | Cat. #: HZ-1296 | 200 ng/ml |
| Peptide, recombinant protein | R-Spondin-1 | Peprotech | Cat. #: 120–38 | 200 ng/ml |
| Peptide, recombinant protein | A83-01 | MCE | Cat. #: HY-10432 | 0.5 uM |
| Peptide, recombinant protein | p38 inhibitor SB202190 | Sigma | Cat. #: SB202190 | 10 uM |
| Peptide, recombinant protein | Human Chorionic Gonadotropin (HCG) | Livzon Pharmaceutical Group Inc | Cat. #: 2000 U | 1 µg/ml |
| Peptide, recombinant protein | Human Placental Lactogen (HPL) | R&D Systems | Cat. #: 5757-PL | 20 ng/ml |
| Peptide, recombinant protein | Prolactin | Peprotech | Cat. #: 100–07 | 20 ng/ml |
| Commercial assay or kit | Periodic acid-Schiff staining | Solarbio | Cat. #: G1280 | |
| Commercial assay or kit | RNeasy Micro Kit | Qiagen | Cat. #: 74004 | |
| Commercial assay or kit | Qubit High Sensitivity RNA Kit | Thermo Fisher Scientific | Cat. #: Q32855 | |
| Commercial assay or kit | gDNA wiper Mix | Vazyme | Cat. #: R323 | |
| Commercial assay or kit | HiScript III qRT SuperMix | Vazyme | Cat. #: R323 | |
| Commercial assay or kit | SYBR Green Premix Pro Taq HS qPCR Kit | Accurate Biotechnology (human) Co., Ltd | Cat. #: AG11701 | |
| Commercial assay or kit | Duolink PLA | Sigma | Cat. #: DUO92008, DUO82049 | |
| Commercial assay or kit | ATP Test Kit | Beyotime | Cat. #: S0026 | |

*Appendix 1 Continued on next page*

*Appendix 1 Continued*

| Reagent type (species) or resource | Designation | Source or reference | Identifiers | Additional information |
|---|---|---|---|---|
| Commercial assay or kit | Human IL-8/CXCL8 ELISA Kit | ABclonal | Cat. #: RK00011 | |
| Chemical compound, drug | Antibiotic-Antimycotic (100 X) | Gibco | Cat. #: 15240062 | 1% |
| Chemical compound, drug | ITS | Gibco | Cat. #: 41400–045 | 1% |
| Chemical compound, drug | L-Glutamine | Gibco | Cat. #: 25030–081 | 2 mM |
| Chemical compound, drug | Nicotinamide | Sigma | Cat. #: N3376 | 1 mM |
| Chemical compound, drug | B27 | Gibco | Cat. #: 17504–044 | 2% |
| Chemical compound, drug | N2 | Gibco | Cat. #: 17502–048 | 1% |
| Chemical compound, drug | N-acetyl-L-cysteine | Sigma | Cat. #: A7250 | 1.25 mM |
| Chemical compound, drug | Estradiol | Sigma | Cat. #: E2758 | 10 nM |
| Chemical compound, drug | Medroxyprogesterone Acetate | Selleck | Cat. #: S2567 | 1 μM |
| Chemical compound, drug | N6,2′-O-dibutyryladenosine 3′,5′-cyclic monophosphate sodium salt (cAMP) | Sigma | Cat. #: D0627 | 1 μM |
| Chemical compound, drug | ITS-X | Gibco | Cat. #: 51500–056 | 1x |
| Chemical compound, drug | β-estradiol | Sigma | Cat. #: E8875 | 8 nM |
| Chemical compound, drug | progesterone | Sigma | Cat. #: P0130 | 200 ng/ml |
| Chemical compound, drug | sodium lactate | Sigma | Cat. #: L7900 | 0.22% |
| Chemical compound, drug | Sodium pyruvate | Sigma | Cat. #: P4562 | 1 mM |
| Chemical compound, drug | Y27632 | Selleck | Cat. #: S1049 | 10 μM |
| Chemical compound, drug | defined fetal bovine serum | Biosera | Cat. #: bs-0003 | 20% |
| Chemical compound, drug | KOSR | Gibco | Cat. #: A3181501 | 30% |
| Software, algorithm | Monocle | Trapnell et al [62] | Version 2.10.1 | |
| Software, algorithm | CellphoneDB | Efremova M et al [43] | Version 4 | |
| Software, algorithm | Imaris x64 | Oxford Instruments | version 10.0.1 | |
| Software, algorithm | Prism 9 | GraphPad | | |
| Other | DAPI | Biotechnology | Cat. #: C1002 | |
| Other | 0.25% trypsin | HyClone | Cat. #: SH30042.01 | |
| Other | Matrigel | Corning | Cat. #: 536231 | |
| Other | collagenase V | Sigma | Cat. #: C-9263 | |
| Other | dispase II | Sigma | Cat. #: D4693 | |
| Other | DNase I | Worthington | Cat. #: LS002139 | |
| Other | 40 μm cell strainer | Corning | Cat. #: 352340 | |

