## [Editor Report · eLife Assessment]

This **important** study reports an endometrial organoid culture system mimicking the window of implantation. The evidence supporting the conclusion drawn is **convincing**. The data will be of interest to embryologists and investigators working on reproductive biology and medicine.

---

## [Referee Report · Reviewer #2 (Public review)]

Zhang et al. have developed an advanced three-dimensional culture system of human endometrial cells, termed a receptive endometrial assembloid, that models the uterine lining during the crucial window of implantation (WOI). During this mid-secretory phase of the menstrual cycle, the endometrium becomes receptive to an embryo, undergoing distinctive changes. In this work, endometrial cells (epithelial glands, stromal cells, and immune cells from patient samples) were grown into spheroid assembloids and treated with a sequence of hormones to mimic the natural cycle. Notably, the authors added pregnancy-related factors (such as hCG and placental lactogen) on top of estrogen and progesterone, pushing the tissue construct into a highly differentiated, receptive state. The resulting WOI assembloid closely resembles a natural receptive endometrium in both structure and function. The cultures form characteristic surface structures like pinopodes and exhibit abundant motile cilia on the epithelial cells, both known hallmarks of the mid-secretory phase. The assembloids also show signs of stromal cell decidualization and an epithelial mesenchymal transition, like process at the implantation interface, reflecting how real endometrial cells prepare for possible embryo invasion.

Although the WOI assembloid represents an important step forward, it still has limitations: the supportive stromal and immune cell populations decrease over time in culture, so only early-passage assembloids retain full complexity. Additionally, the differences between the WOI assembloid and a conventional secretory-phase organoid are more quantitative than absolute; both respond to hormones and develop secretory features, but the WOI assembloid achieves a higher degree of differentiation due to the addition of "pregnancy" signals. Overall, while it's a reinforced model (not an exact replica of the natural endometrium), it provides a valuable in vitro system for implantation studies and testing potential interventions, with opportunities to improve its long-term stability and biological fidelity in the future.

[Editors' note: the authors have responded to the previous round of recommendations.]

---

## [Author Response]

The following is the authors’ response to the previous reviews

**Public Reviews:**

**Reviewer #1 (Public review):**
Summary:This study generated 3D cell constructs from endometrial cell mixtures that were seeded in the Matrigel scaffold. The cell assemblies were treated with hormones to induce a "window of implantation" (WOI) state. Although many bioinformatic analyses point in this direction, there are major concerns that must be addressed.Strengths:The addition of 3 hormones to enhance the WOI state (although not clearly supported in comparison to the secretory state).Comments on revisions:The authors did their best to revise their study according to the Reviewers' comments. However, the study remains unconvincing, incomplete and at the same time still too dense and not focused enough.
**Reviewer #2 (Public review):**
Zhang et al. have developed an advanced three-dimensional culture system of human endometrial cells, termed a receptive endometrial assembloid, that models the uterine lining during the crucial window of implantation (WOI). During this mid-secretory phase of the menstrual cycle, the endometrium becomes receptive to an embryo, undergoing distinctive changes. In this work, endometrial cells (epithelial glands, stromal cells, and immune cells from patient samples) were grown into spheroid assembloids and treated with a sequence of hormones to mimic the natural cycle. Notably, the authors added pregnancy-related factors (such as hCG and placental lactogen) on top of estrogen and progesterone, pushing the tissue construct into a highly differentiated, receptive state. The resulting WOI assembloid closely resembles a natural receptive endometrium in both structure and function. The cultures form characteristic surface structures like pinopodes and exhibit abundant motile cilia on the epithelial cells, both known hallmarks of the mid-secretory phase. The assembloids also show signs of stromal cell decidualization and an epithelial mesenchymal transition, like process at the implantation interface, reflecting how real endometrial cells prepare for possible embryo invasion.Although the WOI assembloid represents an important step forward, it still has limitations: the supportive stromal and immune cell populations decrease over time in culture, so only earlypassage assembloids retain full complexity. Additionally, the differences between the WOI assembloid and a conventional secretory-phase organoid are more quantitative than absolute; both respond to hormones and develop secretory features, but the WOI assembloid achieves a higher degree of differentiation due to the addition of "pregnancy" signals. Overall, while it's a reinforced model (not an exact replica of the natural endometrium), it provides a valuable in vitro system for implantation studies and testing potential interventions, with opportunities to improve its long-term stability and biological fidelity in the future.
**Recommendations for the authors:**

**Reviewer #1 (Recommendations for the authors):**
This study generated 3D cell constructs (i.e., assembloids) that were treated with hormones to induce a 'window of implantation' (WOI) state. While the authors have made large efforts to address the reviewers' feedback, the study's findings remain unconvincing and incomplete.(1) The authors have appropriately revised the terminology from 'organoids' to 'assembloids' in several parts of the manuscript. However, this revision remains incomplete, as the main title, figure legends, and figure titles still contain the incorrect term. A thorough review of the entire manuscript is recommended to ensure consistent and accurate use of terminology.

Thank you for your meticulous review. We have now conducted a full check and confirmed that terminology is used consistently and accurately throughout the text.

(1) Previous comments raised concerns about the feasibility of robustly passaging assembloid structures - comprising epithelial, stromal and immune cells - under epithelial growth conditions. The authors responded by stating that they optimized the expansion medium with a stromal cell-promoting factor. Additionally, rather than conducting scRNA-seq on both early and late passages (P6-P10) as suggested, they performed immunofluorescence staining, which confirmed the persistence of stromal cells at passage 6. However, the presence of immune cells was not addressed. Confirmation of their presence is essential for all further claims. Moreover, a more zoomed-out view of the immunostaining would help clarify the overall cellular composition across the entire well and facilitate comparison with corresponding brightfield images.

Whole-mount immunofluorescence of the 6th - generation assembloids revealed that CD45^+^ immune cells surrounded FOXA2^+^ glands, with a more zoomed-out view provided.

**Author response image 1. sa2fig1:** Whole-mount immunofluorescence showed that CD45^+^ cells (immune cells) were arranged around the glandular spheres that were FOXA2^+^. Scale bar = 50 μm (left) and 30 μm (right).

In their response, the authors mention using the first three passages to ensure optimal cell diversity and viability. However, the manuscript states that 'assembloids derived from the first generation are used for experiments' (line 106). This discrepancy must be clarified.

Thank you for your suggestion. We have revised the relevant content to “The assembloids derived from the first three generation are used for experiments” (Line 90-91).

(2) The authors have made a commendable effort to bring more focus to the manuscript, which has improved readability.

We thank you for your insightful suggestions, which have greatly improved the quality of our manuscript.

(3) The "embryo implantation" part remains very unconvincing. How did authors define "the blastoids could grow within the endometrial assembloids and interact with them"? What did they mean with "grow"? Did blastoids further differentiate? Normally, blastoids cannot further "grow". "Survival rates of blastoids" is not equal to "growth". It is not clear how the survival rate was quantified. Besides, regarding the "interaction rates", how did authors define and quantify it? Actually, blastoids are able to attach to Matrigel efficiently (even without any endometrial cells), so authors cannot simply define the "interaction" as the co-localization of blastoids and assembloids via brightfield images. In addition, for the assembloids as the 3D structures grow in the Matrigel, the epithelial parts are normally apical-in, while the blastoids attach to the apical (lumen) side of the epithelial cells, so physiologically, blastoids should interact with the apical part of the epithelial cells instead of the outside of the assembloids.(1) What did they mean with "grow"? Did blastoids further differentiate?

On the one hand, volume and morphology undergo continuous dynamic changes; on the other hand, only the inner cell mass and trophectoderm exist at the blastocyst stage, with the ICM further differentiating into OCT4^+^ epiblast and GATA6^+^ hypoblast.

(2) Survival rates of blastoids" is not equal to "growth". It is not clear how the survival rate was quantified.

The definition of "survival rate" is as follows: morphologically, the blastocoel remains noncollapsed and the cell boundaries are distinct (with no obvious cell detachment); molecularly, the markers of epiblast, hypoblast and trophectoderm are expressed. The survival rate is calculated as the ratio of viable embryoids to the total number of embryoids.

(3) Besides, regarding the "interaction rates", how did authors define and quantify it? Actually, blastoids are able to attach to Matrigel efficiently (even without any endometrial cells), so authors cannot simply define the "interaction" as the co-localization of blastoids and assembloids via brightfield images.

The criteria for determining interaction include not only attachment between the blastoids and assembloids observed via brightfield images, but also their sustained tight adhesion against external mechanical perturbations (e.g., medium replacement, immunostaining procedures).

(4) In addition, for the assembloids as the 3D structures grow in the Matrigel, the epithelial parts are normally apical-in, while the blastoids attach to the apical (lumen) side of the epithelial cells, so physiologically, blastoids should interact with the apical part of the epithelial cells instead of the outside of the assembloids.

You are absolutely correct. In vivo, the embryo indeed makes initial contact with the apical side of the epithelial cells. The introduction of the blastoid co-culture model herein is intended to demonstrate that this receptive endometrial assembloids can better support blastoid growth and development.

(4) Previous comments highlighted the absence of distinct shifts in gene expression profiles between SEC assembloids and WOI assembloids, which contrasts with findings from primary endometrial tissue reported by Wang et al. (2020). While the authors have expanded their analysis using the Mfuzz algorithm and identified changes in mitochondria- and cilia-associated genes, the manuscript still lacks evidence of significant transcriptional changes in key WOI marker genes, as described in Wang et al. This discrepancy must be addressed and discussed in greater depth to clarify the biological relevance of their model.

The endometrium in vivo involves complex crosstalk among multiple cell types and is tightly regulated by the hypothalamic-pituitary-ovarian (HPO) axis, thus exhibiting distinct shifts in gene expression during the peri-implantation period.

In our in vitro model, alterations in mitochondria- and cilia-related genes were observed, which to a certain extent demonstrates that these window of implantation (WOI) assembloids possess receptive-phase characteristics and can be employed to investigate WOI-associated scientific questions or conduct in vitro drug screening.

However, substantial efforts are still required to optimize the current model for fully recapitulating the dynamic changes in endometrial gene expression across different phases in vivo, and this aspect is further addressed in the Limitations section of our discussion (Line 342-353).

“However, our WOI endometrial assembloids also exhibit some limitations. It is undeniable that the assembloids cannot perfectly replicate the in vivo endometrium, which comprises functional and basal layers with a greater abundance of cell subtypes, under superior regulation by hypothalamic-pituitary-ovarian (HPO) axis. Specifically, stromal and immune cells are challenging to stably passage, and their proportion is lower than in the in vivo endometrium. While the in vivo peri-implantation period exhibits intricate gene expression dynamics driven by systemic regulation, our models only partially recapitulate these changes, primarily in mitochondria- and cilia-associated genes. Nevertheless, to some extent, these WOI assembloids possess receptivity characteristics and can be utilized for investigating receptivity-related scientific questions or conducting in vitro drug screening. Further refinements are required to fully simulate the dynamic endometrial gene expression patterns across all menstrual cycle stages. We are looking forward to integrating stem cell induction, 3D printing, and microfluidic systems to modify the culture environment.”

(5) In the authors' response document, they present data integrating their results with those of Garcia Alonso et al. (2021). However, these integrated analyses are not included in the revised manuscript (which should be, if answering a major concern).

Thanks for your valuable suggestions. We have now integrated the findings of Garcia Alonso et al. (2021) into the revised manuscript (Line 132) and Figure S2E–F.

(8) Fig 2D: The authors have clarified that CD45+ staining is used. However, they have not yet adapted the typo in the figure legend of the right picture.

Thanks for your thorough review. The left panel of Figure 2D is stained with CD45 to label immune cells, while the right panel is stained with CD44. These details have been clearly indicated in both the manuscript and the figure legend.

(9) All quantification analyses (as described in the authors' response document) should be clearly described in the Materials & Methods section.

Thanks for your valuable suggestions. All quantification analyses have now been added to the Supporting Materials and Methods section (Line 94-104, Line 110-111, Line 241244).

(10) The authors have provided clarification regarding their method for quantifying immunofluorescence staining (e.g., OLFM4 expression in Fig. 3C) in their response document. However, these methodological details are not included in the revised manuscript. It is important that such information is incorporated into the manuscript itself to ensure transparency and reproducibility for others.

Thanks for your valuable suggestions. All quantification analyses have now been added to the Supporting Materials and Methods section (Line 94-104).

(13) It is needed to include the author's response to the comment about literature showing the opposite of increased number of cilia during the WOI into the discussion part of the paper.

We appreciate your suggestions. The relevant content has now been added to the Discussion section (Lines 319–323).

(14) In the authors' response, they explain the difference between pinopodes and microvilli. They should include this explanation briefly in the manuscript. Moreover, Fig. 3F lacks a picture of cilia structure in CTRL condition. In addition, the structures that are indicated as cilia with an orange arrow seem to not be attached to the endometrial cells (anymore). It would be useful to show another more representative picture for the cilia.

(1) Thank you for your valuable suggestions. The distinction between pinopodes and microvilli has now been added to the Supporting Materials and Methods section (Line 230-236).

(2) You are probably referring to Figure 2F—we did not observe ciliary structures in the CTRL group.

(3) The cilia structure was visualized via transmission electron microscopy (TEM), which requires ultrathin sectioning. Thus, the cilia shown in the image correspond to a single cross-section of the captured assembloids. Owing to technical limitations, three-dimensional visualization of cilia on the cells cannot be achieved.

(17) The results on co-culturing blastoids with the WOI assembloids is not convincing. The blastoids are exposed to the basolateral side of the endometrial epithelial cells, while in vivo, blastocysts interact with the apical side of the endometrial epithelial cells first (apposition and attachment), followed by invasion into the endometrium. This means that the interaction shown here is not physiological. Therefore, it is not justified to say that this platform holds promise to investigate maternal-fetal interactions.

We agree with your perspective that discrepancies exist between this model and the physiological processes in vivo. However, such differences do not negate the scientific value of the model.

The core merit of this study lies in the successful establishment of co-culture systems for blastoids and WOI assembloids. Notably, genuine cross-talk occurs between the two components, thereby providing a practical and operational tool for subsequent research.

Although the current contact orientation differs from that observed in vivo, future optimization of the cell culture protocol (via modulation of cell polarity) will enable the model to better recapitulate physiological conditions. Therefore, the innovation and operability of this model within specific research contexts still render it a robust platform for investigating maternal-fetal interactions.

Overall, it is highly recommended that the authors carefully review the manuscript for grammatical errors, inconsistencies and issues with scientific phrasing. The language throughout the text requires substantial editing to improve clarity, readability and precision.

We appreciate your suggestions. A full manuscript check was performed to rectify grammatical errors, inconsistencies, and inappropriate scientific phrasing, with further language refinement by a native English-speaking specialist.

Fig 1A: This overview is unclear. How many days do the assembloids grow before being stimulated with hormones? Are CTRL assembloids only kept in culture until day 2 and SEC and WOI assembloids until day 8? This is also not clear form the Materials and Methods section. Should be clarified.

Thanks for your valuable suggestions. We have now updated the overview (Figure 1A) and Materials and Methods section (Line 370-371, Line 379-381).

“Hormonal treatment was initiated following the assembly of the endometrial assembloids (about 7-day growth period).”

“The CTRL group was cultured in ExM without hormone supplementation and subjected to parallel culture for 8 days along with the two aforementioned groups.”

Fig 1B: From these brightfield images, it appears that the size of the assembloids remains relatively consistent from Day 0 to Day 3 and up to Day 11 (especially in CTRL). However, in Fig S1A, the assembloids on Day 11 appear significantly larger compared to those on Day 2 (or Day 4). Authors should clarify this discrepancy (since both of the figures are shown as "brightfield of endometrial assembloids").

You are probably referring to the observation that the assembloids at Day 11 in Fig. S1A are smaller in size than those at Day 2 (or Day 4) in Fig. 1B. This discrepancy arises because the time points in Fig. 1B are calculated starting from the initiation of hormone treatment for the SEC and WOI groups, rather than from the beginning of the overall culture as in Fig. S1A. In addition, assembloids exhibit size variability during the same culture period due to individual heterogeneity.

To eliminate ambiguity, we have now labeled “Hormone Day 0, Day 2, Day 8” in Fig. 1B and revised the corresponding figure legend to read: “Endometrial assembloids from the CTRL, SEC, and WOI groups, which were subjected to hormone treatment on Days 0, 2, and 8, exhibited comparable growth patterns throughout the culture period.”

Fig 2G: authors still used the description "organoids" here instead of "assembloids".

We appreciate your careful review. Corrections have been made accordingly.

Fig. 3C: For the OLFM4 staining quantification, in the Y-axis authors wrote "proportion of OLFM4 (+) cells (OLFM4 (+)/total)", but in the rebuttal letter they mention "its fluorescence intensity (quantified as mean grey value) was significantly stronger in both the SEC and WOI groups compared to the CTRL group". This is confounding and should be clarified.

We apologize for incorrectly writing "fluorescence intensity" in the rebuttal letter; the correct term should be the "proportion of OLFM4 (+) cells (OLFM4 (+)/total)" as shown in Fig. 3C.

Fig 5D: Acetyl-α-tubulin is the marker of ciliated cells and should be expressed in the cilia instead of the whole cells. It is very strange to quantify as "mean fluorescence intensity (acetyl-αtubulin/DAPI)" to assess the cilia. Please clarify.

Thank you for your insightful comment. To clarify, the ratio "mean fluorescence intensity (acetyl-α-tubulin/DAPI)" was calculated within individual acetyl-α-tubulin^+^ ciliated cells. Acetyl-αtubulin fluorescence was normalized to the DAPI signal of the same cell nucleus, not the wholecell population. This corrected for variations in cell number and staining efficiency to ensure data accuracy.

Fig 5F: it is very bizarre that unciliated epithelium was transformed from ciliated epithelium, and CTRL was transformed from SEC and WOI. Should be clarified and discussed.

Pseudotime analysis sorts discrete cells along a "pseudotime axis" based on similarities and differences in cellular gene expression, thereby simulating cell state transitions.

Ciliated epithelium → unciliated epithelium: During the menstrual cycle, ciliated and unciliated epithelia undergo mutual transformation from the secretory phase (or mid-secretory phase) to the menstrual phase, and then to the proliferative phase. Here, we demonstrate the transition of ciliated cells to unciliated cells from the SEC and WOI stages to the CTRL stage.

Notably, the two cell types coexist, and what is presented here merely reflects a transformation trend. Relative content has been incorporated into the Discussion section (Line 319-321).

“Throughout the menstrual cycle, ciliated and unciliated epithelia undergo mutual transformation from the secretory phase (or mid-secretory phase) to the menstrual phase, and then to the proliferative phase.”

Fig 5H: To show "enhanced invasion ability", authors must provide some quantification and statistic analysis. It is very hard to see the difference between the CTRL and SEC regarding ROR2Wnt5A.

We appreciate your suggestion. Quantification and statistic analysis have been added to Figure 5H.

Fig 6A: please elaborate the "mIVC1" and "mIVC2" in the figure legends.

Additions have been made to the figure legends accordingly, as follows: "mIVC1: modified In Vitro Culture Medium 1; mIVC2: modified In Vitro Culture Medium 2."

Fig S1D: Is the PAS staining also done in CTRL assembloids? In addition, it is stated that the assembloids secrete glycogen because of a positive PAS staining, while it could also be neutral mucins, glycoproteins, etc, which are all detected by PAS staining. So, the authors should be more careful in stating that it is glycogen, or a PAS staining with diastase digestion should be done.

The PAS staining results for the CTRL group are presented in Fig. S1I. In addition, results of PAS staining with diastase digestion are included in Figure S1.

Line 120: references?

The reference has been added accordingly.

Line 178: The term 'Endometrial Receptivity Test (ERT)' is used. Do the authors mean Endometrial Receptivity Analysis (ERA) test? ERA is the commonly used abbreviation for this test. Moreover, the authors describe ERA as 'a kind of gene analysis-based test.' This should be rephrased more scientifically correct.

Thank you for your valuable suggestion. We have revised the term to ERA, and modified the phrase "a kind of gene analysis-based test" to "gene expression profiling-based diagnostic assay" (Lines 160–163).

“We performed Endometrial Receptivity Analysis (ERA), a gene expression profiling-based diagnostic assay that integrates high-throughput sequencing and machine learning to quantify the expression of endometrial receptivity-associated genes.”

Line 83: assemblies à assembloids

We appreciate your suggestion. The text has been updated to “the endometrial assembloids progressed from epithelial organoids, to assemblies of epithelial and stromal cells and then to stem cell-laden 3D artificial endometrium”.

The Materials and Methods section currently lacks the needed details. Authors should substantially expand this section to clearly describe all experimental and analytical procedures, including, aùmong others, immunofluorescence staining, quantification methods, bioinformatics analyses and statistical approaches. Providing comprehensive methodological information is essential.

A detailed description of these methods is provided in the Supporting Materials and Methods section.

**Reviewer #2 (Recommendations for the authors):**
The revised manuscript is much improved in clarity, focus, and experimental support. The authors have thoughtfully addressed the major concerns from the previous review. In particular, the logic and flow of the paper are clearer, it now guides the reader through the rationale (constructing a WOI model), the comparative analysis against in vivo tissue and simpler organoids, and the key features that distinguish the WOI assembloid. The added functional validation (especially the blastoid co-culture experiment) significantly strengthens the work by showing a tangible outcome of "receptivity" beyond molecular profiling. The distinction between the standard secretory-phase organoid and the WOI assembloid is now more convincing, as the authors highlight several specific differences in morphology (more cilia, pinopodes), metabolism, and implantation success that favor the WOI model. The manuscript also reads cleaner with the bioinformatic sections condensed to the most important findings (excess detail was trimmed or moved to supplements) and the rationale for gene/pathway selection explicitly stated.The manuscript has been significantly strengthened through the addition of functional assays (like the blastoid co-culture), clearer transcriptomic and proteomic data, and detailed analyses of hormone treatments, cilia biology, and stromal and immune cell behavior in early passages. These updates confirm that the WOI assembloid supports embryo attachment and outperforms standard secretory organoids, while integrating external references and clarifications on terminology. Minor suggestions remain, such as clarifying statistical significance and adding functional interpretations for certain observations, but overall, the manuscript is now more robust and biologically convincing.Remaining points for clarification: There are a few minor points that still merit attention:- Use of the Endometrial Receptivity Test (ERT): As previously mentioned, if the authors have ERT data for the SEC organoid group, including that information would further support the claim that the WOI assembloid is uniquely receptive. If not, it would be helpful to add a statement clarifying that the ERT was employed specifically as a confirmatory test for the WOI assembloids, rather than as a comparative measure across all groups.

Thank you for your valuable suggestion. We have now supplemented the description in the Supporting Materials and Methods section (Lines 160–162) as follows: “ERA was employed specifically as a confirmatory test for the WOI assembloids, rather than as a comparative measure across all groups.”

- Because the assembloids are created from primary tissue samples, it would be helpful to briefly comment on how consistent the findings were across different patient-derived samples. For example, did all biological replicates show similar expression of receptivity markers and comparable capacity to support blastoid attachment? Although this seems implied, including a sentence in the Methods or Results sections that specifies the number of donor lines tested would help readers assess the model's variability and reproducibility.

We appreciated your advice. The relevant statement has been added to the Supporting Materials and Methods section. (Line 312-313).

“All biological replicates (fourteen individuals) of endometrial assembloids show similar expression of receptivity markers and comparable capacity to support blastoid attachment.”

- The authors mention promising future directions, such as integrating 3D printing and microfluidics to further enhance the model, which is an excellent forward-looking statement. It would also be valuable to suggest the inclusion of additional cell types, like more robust immune cell populations or endothelial components, as future improvements to create an even more comprehensive model of the endometrial lining.

Thank you for your valuable suggestion. 3D printing and microfluidics serve as approaches for introducing multiple cell types. We have supplemented the following statement in the manuscript: “We are looking forward to integrating stem cell induction, 3D printing, and microfluidic systems to modify the culture environment.” (Line 352-353).

We are grateful for your valuable feedback and constructive criticism, which have helped us improve the quality of our work in terms of content and presentation. We have diligently revised the manuscript and made necessary changes. Here, we have attached the revised manuscript, figures, and all supplementary materials for your re-evaluation. Thank you again for your continued support and look forward to your favorable decision.